# MALAT1-dependent hsa_circ_0076611 regulates translation rate in triple-negative breast cancer

Chiara Turco[1], Gabriella Esposito[1], Alessia Iaiza[2], Frauke Goeman[3], Anna Benedetti[2], Enzo Gallo [4],
Theodora Daralioti[4], Letizia Perracchio[4], Andrea Sacconi[5], Patrizia Pasanisi[6], Paola Muti[7,8], Claudio Pulito [1],
Sabrina Strano[3], Zaira Ianniello[9], Alessandro Fatica [9], Mattia Forcato [10], Francesco Fazi [2✉],
Giovanni Blandino [1✉] & Giulia Fontemaggi [1✉]

Vascular Endothelial Growth Factor A (VEGFA) is the most commonly expressed angiogenic growth factor in solid tumors and is generated as multiple isoforms through alternative mRNA splicing. Here, we show that lncRNA *MALAT1* (metastasis-associated lung adeno-carcinoma transcript 1) and ID4 (inhibitor of DNA-binding 4) protein, previously referred to as regulators of linear isoforms of *VEGFA*, induce back-splicing of *VEGFA* exon 7, producing circular RNA circ_0076611. Circ_0076611 is detectable in triple-negative breast cancer (TNBC) cells and tissues, in exosomes released from TNBC cells and in the serum of breast cancer patients. Circ_0076611 interacts with a variety of proliferation-related transcripts, included *MYC* and *VEGFA* mRNAs, and increases cell proliferation and migration of TNBC cells. Mechanistically, circ_0076611 favors the expression of its target mRNAs by facilitating their interaction with components of the translation initiation machinery. These results add further complexity to the multiple *VEGFA* isoforms expressed in cancer cells and highlight the relevance of post-transcriptional regulation of *VEGFA* expression in TNBC cells.

[1] Oncogenomic and Epigenetic Unit, IRCCS Regina Elena National Cancer Institute, Rome, Italy. [2] Department of Anatomical, Histological, Forensic and Orthopedic Sciences, Section of Histology and Medical Embryology, Sapienza University of Rome, Laboratory Affiliated to Istituto Pasteur Italia-Fondazione Cenci Bolognetti, Rome, Italy. [3] UOSD SAFU, Department of Research, Advanced Diagnostics, and Technological Innovation, Translational Research Area, IRCCS Regina Elena National Cancer Institute, Rome, Italy. [4] Department of Pathology, IRCCS Regina Elena National Cancer Institute, Rome, Italy. [5] UOSD Clinical Trial Center, Biostatistics and Bioinformatics, IRCCS Regina Elena National Cancer Institute, Rome, Italy. [6] Unit of Epidemiology and Prevention, Fondazione IRCCS Istituto Nazionale dei Tumori, Milan, Italy. [7] Department of Health Research Methods, Evidence, and Impact, Faculty of Health Sciences, McMaster University, Hamilton, ON, Canada. [8] Department of Biomedical, Surgical and Dental Sciences, "Università degli Studi di Milano", Milan, Italy. [9] Department of Biology and Biotechnology "Charles Darwin", Sapienza University of Rome, Rome, Italy. [10] Department of Life Sciences, University of Modena and Reggio Emilia, Modena, Italy. ✉email: francesco.fazi@uniroma1.it; giovanni.blandino@ifo.it; giulia.fontemaggi@ifo.it

Vascular endothelial growth factor A (VEGFA), also known as VEGF, was initially identified as an endothelial cell-specific mitogen that has the capacity to induce physiological and pathological angiogenesis, lymphangiogenesis and vascular permeability[1,2]. VEGFA is the principal angiogenic promoter in most, if not all, cancers acting primarily on endothelial cells through its cognate receptors VEGF-R1 and VEGF-R2[3]. It has become apparent that the function of VEGFA is not limited to angiogenesis. VEGFA, secreted by tumor and stromal cells, has indeed been shown to impact on cancer biology in various ways, through both autocrine and paracrine mechanisms. For example, some tumor cells express receptors for VEGFA, such as VEGF-R2 and neuropilins, and VEGFA signaling in these cells is associated with aggressive behavior, including the acquisition of stem-like traits[4,5]. Concomitantly, VEGFA can also affect the function of immune cells by recruiting regulatory T cells (Tregs) that inhibit an antitumour immune response[6].

Alternative splicing of VEGFA leads to several different isoforms, generically indicated as VEGF$_{XXX}$, while specifically named VEGF$_{121}$, VEGF$_{145}$, VEGF$_{165}$ and VEGF$_{189}$ in humans, which are differentially expressed in various tumor types[7]. The two major isoforms expressed in cancer cells, VEGF$_{165}$ and VEGF$_{121}$, differ by the inclusion or not of exon 7, whose lack renders VEGF$_{121}$ isoform more soluble and less cell- or matrix-associated[8,9]. Moreover, VEGFA isoforms may be also expressed as VEGF$_{XXX}$b isoforms, generated by use of a distal splice site into last exon 8, which causes an alternate six amino acids sequence at the C terminus, associated to reduced angiogenic potential[7,10,11].

We previously showed that alternative splicing of VEGFA isoforms is modulated, in breast cancer (BC) cells carrying a mutated TP53 gene, by a ribonucleoprotein (RNP) complex enclosing long non-coding RNA MALAT1 (metastasis-associated lung adenocarcinoma transcript 1), splicing factor SRSF1 (Serine And Arginine Rich Splicing Factor 1), ID4 (inhibitor of DNA-binding 4) protein and mutant p53 protein[12,13]. Specifically, this complex modulates VEGFA splicing thanks to the ability of MALAT1 and SRSF1 to interact with VEGFA pre-mRNA, leading to increased synthesis of VEGF$_{XXX}$ vs. VEGF$_{XXX}$b as well as of VEGF$_{121}$ vs. VEGF$_{165}$ isoforms[12]. Modulation of VEGFA expression by ID4 protein in BC cells is also responsible for the reprogramming of tumor-associated macrophages (TAMs). ID4-dependent release of VEGFA from cancer cells indeed induces a paracrine activation of proangiogenic genes, such as for example Granulin, and downregulation of miR-15/107 family members in TAMs[14–16]. lncRNA MALAT1 and ID4 protein are both involved in the promotion of BC aggressive phenotype[17–21]. MALAT1 localizes to nuclear speckles and interacts with a variety of pre-mRNA splicing factors, as for example oncogenic splicing factor SRSF1, modulating their subnuclear distribution and regulating gene expression and splicing[22,23]. ID4 is a helix-loop-helix protein, which specifically marks triple-negative breast cancers (TNBC)[24,25], where it associates with stem-like phenotype and poor prognosis[26].

We here show that the RNP complex enclosing MALAT1 and ID4 controls not only the linear isoforms of VEGFA, but also a circular RNA, formed from back-splicing of VEGFA exon 7, namely hsa_circ_0076611, highlighting a novel mechanism of action of this complex in TNBC cells. Circular RNAs are extremely stable RNA molecules, characterized by covalently closed cyclic structure lacking poly-adenylated tails, and are capable of regulating gene expression at transcriptional and post-transcriptional level. A variety of deregulated circRNAs have been identified in TNBC, some of which also correlate with clinicopathological features and prognosis[27]. We here addressed the mechanisms regulating circ_0076611 expression in TNBC and the functional output of circ_0076611 modulation,

highlighting that this circRNA is able to impinge on the expression, among others, of its parental gene, VEGFA.

## Results

### Circ_0076611 is synthesized from back-splicing of VEGFA exon 7 in TNBC.
The two major isoforms of VEGFA, namely VEGF$_{165}$ and VEGF$_{121}$, differ in the inclusion or not of exon 7 (Fig. 1a), which is reported in circBase (www.circbase.org) as capable of forming a circular RNA product of 132 bp, namely hsa_circ_0076611 (referred to as circ_0076611 hereafter). We first evaluated whether circ_0076611 was actually expressed in BC cells. Validation of circular RNAs is based on the use of divergent primers in RT-PCR, which are expected to give a PCR product only if the analyzed exon is included in a circular RNA in the presence of cDNA as template[28]. Conversely, divergent primers are not expected to give a PCR product when genomic DNA (gDNA) is used as template. By using convergent and divergent primers specific for VEGFA exon 7 in RT-PCR, we observed that convergent primers led to the formation of a PCR product in the presence of both gDNA and cDNA, while divergent primers allowed the formation of a PCR product only in the presence of cDNA as template (Fig. 1b), confirming the existence of circ_0076611 in MDA-MB-468 and HCC1954 cells. Sanger sequencing of PCR product confirmed the presence of the exon 7 back-splice junction (Fig. 1b). To detect circ_0076611 by RT-qPCR, a TaqMan assay containing a probe covering the back-splice junction was developed. Specificity of circ_0076611 detection was assessed by treating or not total RNA from MDA-MB-468 cells initially with RNaseR, degrading specifically the linear forms of RNA, and secondly with RNaseH, degrading RNA/DNA hybrids in presence of an oligonucleotide complementary to VEGFA exon 7 (Fig. 1c, d). Efficacy and specificity of the RNases was demonstrated by the analysis of GAPDH mRNA levels; GAPDH expression indeed was affected only by RNaseR treatment and not by RNaseH (Fig. 1c, d). As expected, decreased circ_0076611 expression was detected in presence of RNaseH, due to the use of an oligonucleotide complementary to VEGFA exon 7, and not in presence of RNaseR. Conversely, the linear VEGFA mRNA (VEGF$_{165}$, enclosing exon 7) was decreased by both RNaseH and RNaseR (Fig. 1c, d). Analysis of a panel of TNBC cell lines revealed different levels of expression of circ_0076611, with MDA-MB-468, HCC1143, HCC70 and HCC1954 showing the highest levels (Fig. 1e). Moreover, profiling of TNBC cell lines indicated a higher expression of circ_0076611 in Basal-A vs. -B molecular subtype[29,30] (Fig. 1f). Finally, we chose two cell lines for their efficient ability to form mammospheres, HCC70 and HCC1954, and we observed greater expression of circ_0076611 in mammospheres than its adherent counterpart (Fig. 1g).

We next verified whether circ_0076611 was expressed in BC tissues. To this end, we initially assessed by RT-qPCR that circ_0076611 was detectable in 6 out of 12 tumor tissues and in none of the matched adjacent tissues from estrogen receptor-negative BC patients (Fig. 2a). Secondly, we developed an in situ hybridization (ISH) assay using the BaseScope technology to detect circ_0076611. BaseScope ISH is specifically designed to detect splicing junctions and is based on the use of a double probe set, covering the 5' and 3' splice junction, respectively, that emit a signal only when they are closely associated, and therefore in the presence of splicing. ISH analysis of a tissue microarray (TMA) containing TNBC tissues revealed that circ_0076611 presents a nuclear or perinuclear localization (Fig. 2b) and is detectable in nearly half of the cases, both considering the whole TMA and the subgroup of cytokeratin-5 positive cases (Ck5$^+$), which distinguishes the basal-like tumors (BLBC) (Fig. 2c). We observed a

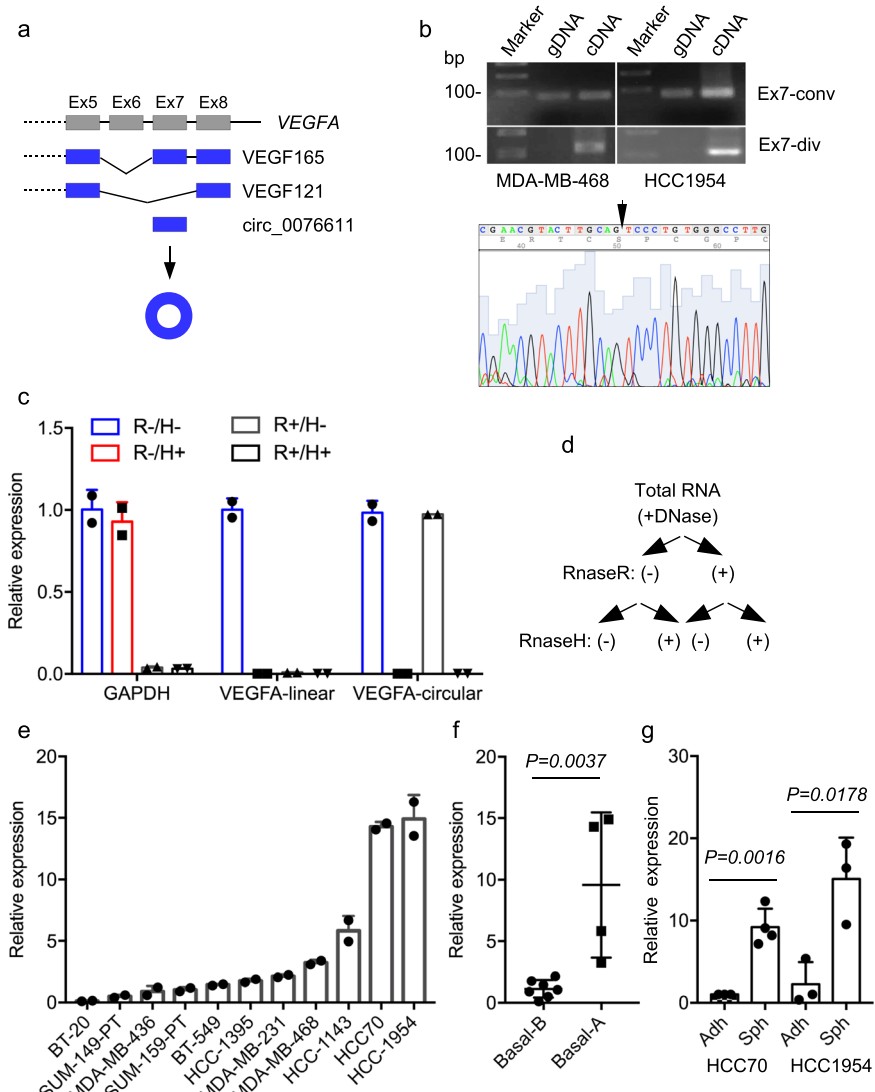

**Fig. 1 Circ_0076611 is expressed in triple-negative breast cancer cell lines. a** Schematic illustration of VEGFA exons 5 to 8. Inclusion or exclusion of exon 7 in alternative splicing events causes to the formation of the two major isoforms, VEGF165 or VEGF121, respectively. Back-splicing of Exon 7 leads to the formation of a circular RNA, circ_0076611. **b** RT-PCR analysis of VEGFA Exon 7 using convergent (Ex7-conv) or divergent (Ex7-div) primers and genomic DNA (gDNA) or cDNA as templates in the indicated cell lines. The presence of circ_0076611 back-splice junction sequence has been verified by Sanger sequencing of PCR products (lower panel). **c**, **d** RT-qPCR analysis of GAPDH, VEGFA linear (VEGF165) and VEGFA circular (circ_0076611) transcripts on total RNA extracted from MDA-MB-468 cells and treated or not with RNaseR and/or RNaseH, in the presence of a VEGFA Exon 7 antisense oligonucleotide. Bars indicate standard error. **e** RT-qPCR analysis of circ_0076611 in the indicated breast cancer cell lines. Bars indicate standard error. **f** Box plot representing the expression of circ_0076611, evaluated by RT-qPCR, in breast cancer cell lines belonging to Basal A and Basal B molecular subtypes. *$p \leq 0.05$ (unpaired, two-tailed Student's *t* test). **g** Expression of circ_0076611, evaluated by RT-qPCR, in the indicated breast cancer cell lines grown in adhesion (Adh) or as mammospheres (Sph). Data are presented as mean plus standard deviation. *$p \leq 0.05$, **$p \leq 0.005$ (unpaired, two-tailed Student's *t* test). Results from at least three biological replicates are shown. Bars indicate standard deviation.

significant association between the presence of circ_0076611 and tumor size in TNBC (Fig. 2d). Moreover, we observed that the more advanced tumor stages showed a greater number of circ_0076611-positive cases (Fig. 2e).

Expression in BC tissues prompted us to investigate whether circ_0076611 is also detectable in secretory samples such as exosomes isolated from conditioned media (CM) of BC cell lines and serum samples from BC patients. As shown in Fig. 2f, g, we observed an enrichment of circ_0076611 level in exosomes from CM of two TNBC cell lines (MDA-MB-468 and HCC1395, representative of the basal A and B subtypes, respectively) and one ovarian cancer cell line (Ovcar-3) compared to their adherent cellular counterparts. Circ_0076611 was detectable also in serum

samples from BC patients (Fig. 2h). As these serum samples were collected for previous studies in the context of a randomized controlled clinical trial designed to evaluate the effect of Metformin on the serum level of hormones (such as testosterone, insulin, androgen and estrogen) in non-diabetic women with BC[31–33], we decided to evaluate also if circulating circ_0076611 levels were changed in the available serum samples collected after treatment with Metformin. Recent studies indeed revealed that Metformin, a primary anti-diabetic agent, shows anti-tumorigenic effects, such as cell cycle arrest and apoptosis, also in BC cells[34]. Of note, we observed that treatment with Metformin significantly downregulated serum circ_0076611 levels in those patients showing high basal levels of the circRNA (Fig. 2i).

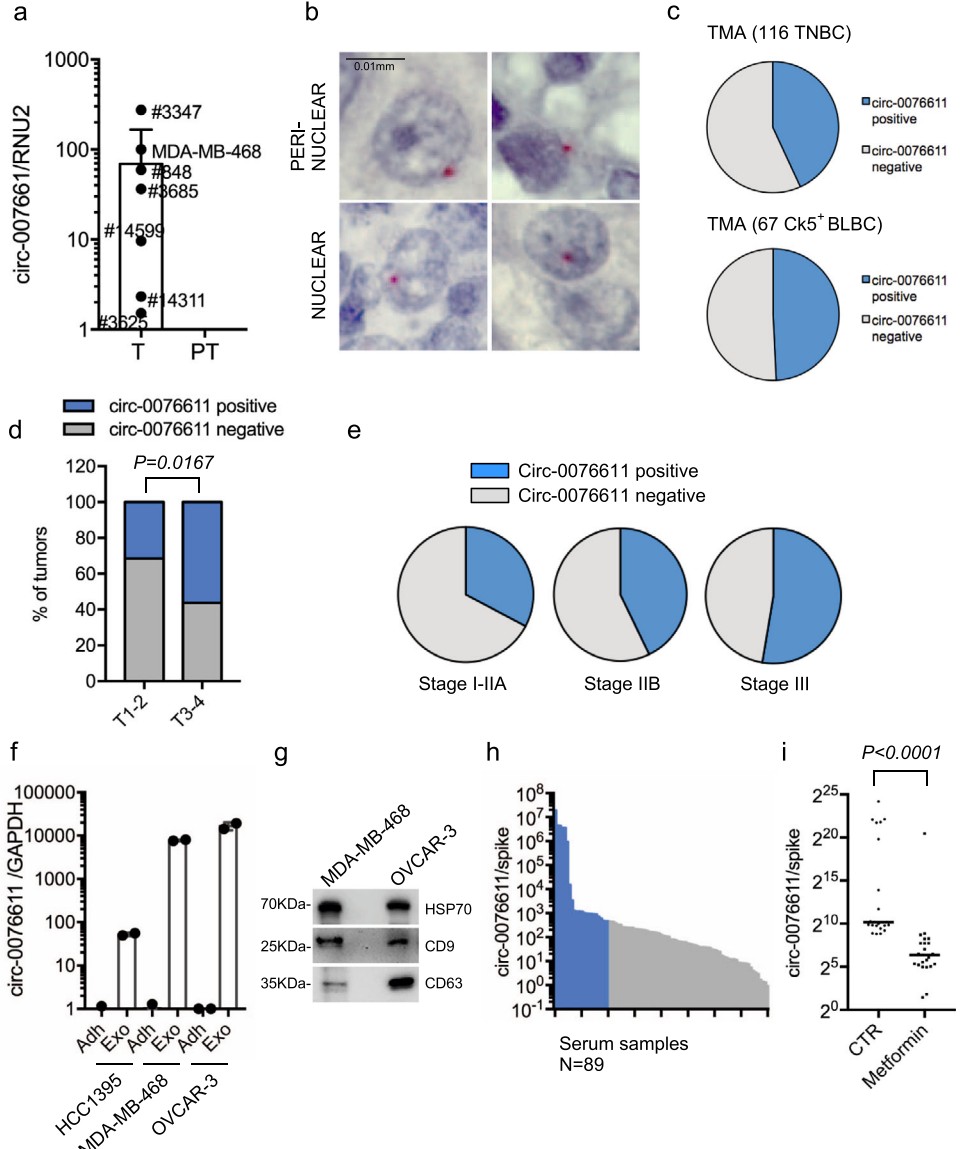

**Fig. 2 Circ_0076611 is expressed in triple-negative breast cancer tissues. a** RT-qPCR analysis of circ_0076611 in fresh frozen ER-negative breast cancer tumor (T) and matched adjacent peritumor (PT) tissues. **b** Detection of circ_0076611 by in situ hybridization (ISH) in breast cancer sections of a tissue microarray (TMA) containing triple-negative breast cancer (TNBC) cases. Scale bar 0.01 mm. **c** Distribution of circ_0076611 expression, as evaluated by ISH, in 116 TNBC tissues and in a subgroup of 67 Cytokeratin-5 positive (Ck5+) sections from the tissue microarray. **d** Percentages of circ_0076611-positive and -negative tumors from the TMA characterized by different size (T1–T2 or T3–T4) ($N = 105$). Two-tailed $p$ value has been calculated by Chi-square test comparing circ_0076611-negative vs. circ_0076611-positive cases, in the two groups with different tumor size T1–T2 and T3–T4. **e** Percentages of circ_0076611-positive and -negative tumors in groups with different tumor stage. **f, g** Expression of circ_0076611 ($\log_{10}$), evaluated by RT-qPCR, in the indicated cell lines (Adh) and in matched exosomes isolated from their conditioned media (Exo) (**f**). Western blot analysis of exosomal markers HSP70, CD9, CD63 has been carried out as quality control for exosomes purification (**g**). Bars indicate standard error. **h** Expression of circ_0076611 ($\log_{10}$), evaluated by RT-qPCR, in the sera of women with established breast cancer ($N = 89$). Samples belonging to the upper quartile are indicated in blue. **i** Scatter plot showing the level of circ_0076611, evaluated by RT-qPCR, in the sera of patients with established breast cancer collected before (indicated as CTR) and after (indicated as Metformin) a 3-months daily treatment with metformin (500 mg day$^{-1}$) as agent with anticancer activity within a randomized clinical trial[31]. The presented data are from patients ($N = 23$) belonging to the upper quartile of the cohort, indicated in blue in **h**. ****$p < 0.0001$. $p$ value has been calculated using Wilcoxon matched-pairs signed rank test.

**Circ_0076611 affects the expression of proliferation-related genes**. Next, we performed a ChIRP assay to identify mRNAs bound by circ_0076611, which may possibly mediate its function, in MDA-MB-468, a TNBC cell line of basal-A subtype previously used in our studies focusing on linear isoforms of VEGFA[12]. ChIRP was performed using a biotinylated antisense oligonucleotide (ASO), which recognizes specifically the back-splice junction of circularized VEGFA exon 7. RNAseq analysis of

RNAs recovered in ChIRP allowed identifying a panel of 321 circ_0076611-interacting mRNAs (Fig. 3a, b). Analysis of the signal distribution showed that circ_0076611 mainly interacts with the coding sequence of its target mRNAs (Supplementary Fig. 1a, b). Various cancer-related pathways were enriched in this panel, being "Protein processing in the ER", "EGFR" and "Hippo signaling" pathways the most significant (Table 1). Of note, among circ_0076611-interacting mRNAs there were several

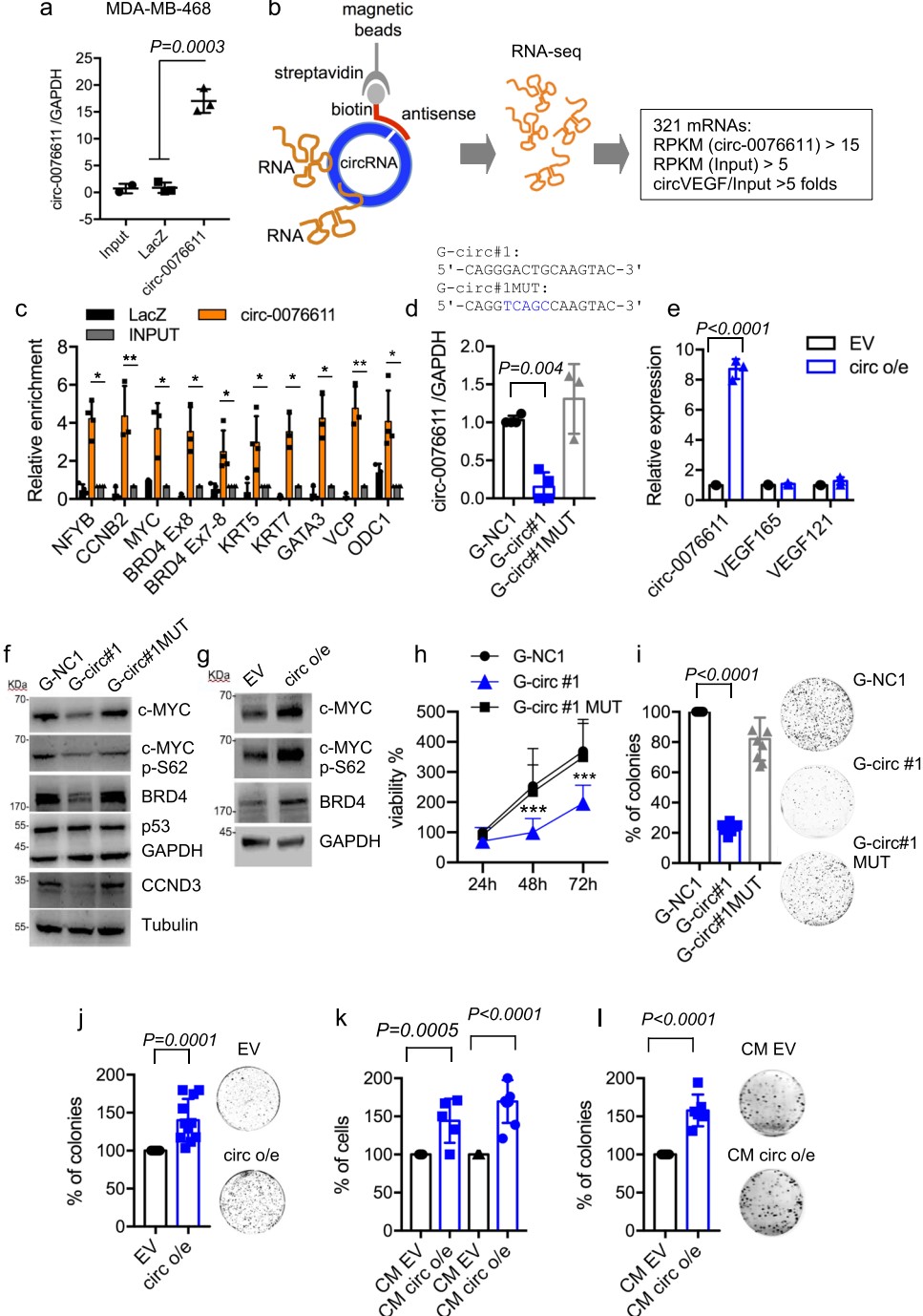

proliferation- and differentiation-related genes (NF-YB, CCNB2, MYC, BRD4, KRT5, KRT7, GATA3, VCP and ODC1), which were validated in independent ChIRP experiments in MDA-MB-468 and in HCC1954, another basal-type-A cell line (Fig. 3c and Supplementary Fig. 1c). To evaluate if modulation of circ_0076611 affects the expression of its interacting mRNAs, we set up circ_0076611 silencing and overexpression in MDA-MB-468 cells. Silencing of circ_0076611 was performed using a GapmeR LNA ASO directed to the back-splice junction (indicated as G-circ #1) or a control oligonucleotide containing 5 mutated nt (indicated as G-circ MUT) or an unrelated negative control oligonucleotide (G-NC1) (Fig. 3d). Overexpression of circ_0076611 was obtained by stable transfection of an expression vector containing VEGFA Exon 7 and part of flanking intronic sequences (Fig. 3e). Of note, silencing of circ_0076611 reduced

(Fig. 3f) while overexpression induced (Fig. 3g) the expression of c-Myc, c-Myc-pSer62 and BRD4, major cell cycle regulatory proteins. Accordingly, upon silencing of circ_0076611 also cell viability (Fig. 3h and Supplementary Fig. 2a, b) and colony-forming ability (Fig. 3i) were reduced. Conversely, circ_0076611-overexpressing cells showed increased colony formation (Fig. 3j). Silencing of circ_0076611 also affected the motility of BC cells, as assessed by transwell and wound-healing assays (Supplementary Fig. 2c–e).

As circ_0076611 is released from TNBC cells in exosomes, we next evaluated the ability of the conditioned medium (CM) from circ_0076611-overexpressing (circ o/e) and control (EV) cells to affect cell proliferation. Of note, both MDA-MB-468 and untransformed mammary epithelial MCF10A cells showed increased proliferation when grown in the presence of CM from

**Fig. 3 Circ_0076611 impacts on cell proliferation. a** RT-qPCR analysis of circ_0076611 in ChIRP assays. Circ_0076611 has been recovered by using a biotinylated antisense oligonucleotide covering the back-splice junction, or negative control LacZ oligonucleotides. circ_0076611 expression has been normalized to GAPDH ($N = 3$ independent biological replicates). $p$ value has been calculated by unpaired, two-tailed Student's $t$ test. **b** Schematic illustration of the identification of circ_0076611-interacting mRNAs by ChIRP-RNAseq. ChIRP assay was performed to recover circ_0076611 and its associated RNAs by using a biotinylated oligonucleotide complementary to circ_0076611 back-splice junction in MDA-MB-468 cells crosslinked with glutaraldehyde. Recovered RNAs were analyzed by RNAseq allowing the identification of 321 target mRNAs, considering the following settings: RPKM (circ_0076611) > 15; RPKM (input) > 5; folds circ_0076611/input > 5. **c** Validation of a panel of circ_0076611-mRNA interactions identified by ChIRP-RNAseq. RT-qPCR analysis of a subset of circ_0076611 target mRNAs has been carried out on RNAs recovered by ChIRP assay in MDA-MB-468 using an oligonucleotide complementary to circ_0076611 back-splice junction or control LacZ oligonucleotides. Enrichments are normalized to GAPDH mRNA and expressed as $\log_2$. *$p \leq 0.05$, **$p \leq 0.005$ (paired, two-tailed Student's $t$ test). Results from three biological replicates are shown. **d** Expression of circ_0076611, evaluated by RT-qPCR, in MDA-MB-468 cells transfected with circ_0076611 GapmeR oligonucleotide (G-circ#1) or control GapmeR G-circ#1 MUT, carrying a 5-nt substitution, or control GapmeR G-NC1. **e** Expression of circ_0076611, VEGF165 and VEGF121, evaluated by RT-qPCR, in MDA-MB-468 cells stably transfected with circ_0076611 expression vector (circ o/e) or an empty vector (EV). **f, g** Western blot analysis of the indicated circ_0076611 targets c-MYC, c-MYC-pSer62 and BRD4, as well as of cell cycle-related cyclin D3 (CCND3) and mutant p53 proteins in MDA-MB-468 transfected as in **d, e**. **h** Cell viability evaluated by ATPlite assay in MDA-MB-468 cell line after transfection of GapmeR oligonucleotide for circ_0076611 silencing (G-circ#1) or control GapmeR G-circ#1 MUT, carrying a 5-nt substitution, or control unrelated GapmeR G-NC1. ***$p \leq 0.0005$ (paired, two-tailed Student's $t$ test) ($N = 3$). **i, j** Colony-formation assay of MDA-MB-468 transfected as in **d, e**. (paired, two-tailed Student's $t$ test) ($N \geq 4$ independent biological replicates). **k, l** Cell counts (**k**) from MCF10A and MDA-MB-468 cells cultured with conditioned medium from circ_0076611-overexpressing MDA-MB-468 (CM circ o/e) or control cells (CM EV, empty vector) for 48 h. Colony-formation assay (**l**) using MDA-MB-468 cells cultured with conditioned medium from circ_0076611-overexpressing MDA-MB-468 (CM circ o/e) or control cells (CM EV, empty vector). CM was replaced every 3 days. **$p \leq 0.005$ (paired, two-tailed Student's $t$ test) ($N \geq 4$ independent biological replicates). Bars indicate standard deviation.

**Table 1 Pathway enrichment analysis of the 321 circ_0076611-interacting mRNAs identified by ChIRP-RNAseq in MDA-MB-468 cells performed by interrogating the ConsensusPathDB tool (http://cpdb.molgen.mpg.de) ($p < 0.05$, $N \geq 3$).**

| $p$ value | Pathway | Source | Genes |
|---|---|---|---|
| 0.0001 | Protein processing in endoplasmic reticulum | KEGG | RAD23B; UBE2J2; UBQLN2; PDIA3; VCP; HSPBP1; UBE2G2; CANX; ATF6B; MAN1A2; NSFL1C |
| 0.0002 | EGFR1 | NetPath | AP2S1; CLTC; SOS1; HGS; GRB10; VAV3; CEBPA; RAP1A; ZPR1; ENO1; CLTA; KRT7; KRT5; INPPL1; ABL1; WBP2; APLP2; MYC; SCAMP3 |
| 0.0012 | Hippo signaling pathway | KEGG | RASSF1; WWC1; NF2; SAV1 |
| 0.0040 | One carbon pool by folate | KEGG | ALDH1L2; GART; ATIC |
| 0.0080 | Parkinson disease | KEGG | TXN2; UBE2J2; UBE2L6; MFN2; NDUFS4; UBE2G2; VDAC2; ATP5MC3; SDHB; UQCRFS1 |
| 0.0110 | Endocrine and other factor-regulated calcium reabsorption | KEGG | AP2S1; CLTA; CLTC; ATP1B1 |
| 0.0153 | Ribosome | KEGG | MRPL34; RPL38; RPL7; RPLP2; MT-RNR1; RPL23; MRPS6 |
| 0.0165 | SNARE interactions in vesicular transport | KEGG | STX5; YKT6; STX6 |
| 0.0208 | DNA replication | KEGG | RFC5; RFC2; PRIM2 |
| 0.0265 | Renal cell carcinoma | KEGG | PAK4; EPAS1; SOS1; RAP1A |
| 0.0294 | Huntington disease | KEGG | AP2S1; POLR2E; CLTA; CLTC; VDAC2; NDUFS4; ATP5MC3; SDHB; UQCRFS1; NRBF2 |
| 0.0331 | Fibroblast growth factor-1 | NetPath | INPPL1; HGS; SOS1; SCAMP3 |
| 0.0374 | Ras signaling pathway | KEGG | SOS1; RASSF1; AFDN; EFNA1; PAK4; SHOC2; RAP1A; ABL1 |
| 0.0380 | Spliceosome | KEGG | SNRPD3; RP9; HNRNPA3; PRPF31; SRSF9; SF3B5 |
| 0.0390 | Antigen processing and presentation | KEGG | NFYB; PDIA3; CANX; HSPA4 |
| 0.0415 | Nucleotide excision repair | KEGG | RFC5; RAD23B; RFC2 |

circ_0076611-overexpressing vs. CM from control cells (Fig. 3k). CM from circ_0076611-overexpressing cells enabled also increased colony-formation of MDA-MB-468, compared to CM from control cells (Fig. 3l). These results highlight the ability of circ_0076611 to induce the expression of proliferation-related genes and to positively modulate proliferation in TNBC cells.

**Circ_0076611 affects the expression of proangiogenic cytokines**. ChIRP-RNAseq intriguingly revealed that circ_0076611 interacts with the last exon of VEGFA (exon 8) (Fig. 4a and Supplementary Fig. 1d), suggesting a possible involvement of this circRNA in the control of the expression of its parental gene, as already reported for numerous circRNAs[35]. Two additional mRNAs encoding for proangiogenic cytokines (CXCL16 and CXCL1) interacted with circ_0076611 in ChIRP-RNAseq (Fig. 4a). Validation ChIRP experiments confirmed the existence of these interactions (Fig. 4b–d and Supplementary Fig. 1e). Of note, silencing of

circ_0076611 markedly reduced the expression of VEGFA proteins, both in the cells (Fig. 4e) and in secreted exosomes (Fig. 4f). Conversely, VEGFA transcripts were not reduced (Fig. 4h, i), while VEGF121 was even slightly increased, after circ_0076611 silencing (Fig. 4h, i), suggesting that circ_0076611 controls VEGFA expression post-transcriptionally. CXCL16 behaved similarly to VEGFA (Fig. 4j, k), while CXCL1 showed significant protein and mRNA reduction following circ_0076611 depletion (Fig. 4l, m), suggesting that CXCL1 might be controlled also at transcriptional level by some circ_0076611-dependent factor, as for example c-Myc.

**Circ_0076611 controls the translation rate of its target mRNAs.** In light of the marked effect of circ_0076611 silencing on the protein expression of its target mRNAs, we aimed at understanding how circ_0076611 impacts on gene expression. Subcellular localization analysis by ISH showed that circ_0076611 is mainly localized in the perinuclear region and in the nucleoli in

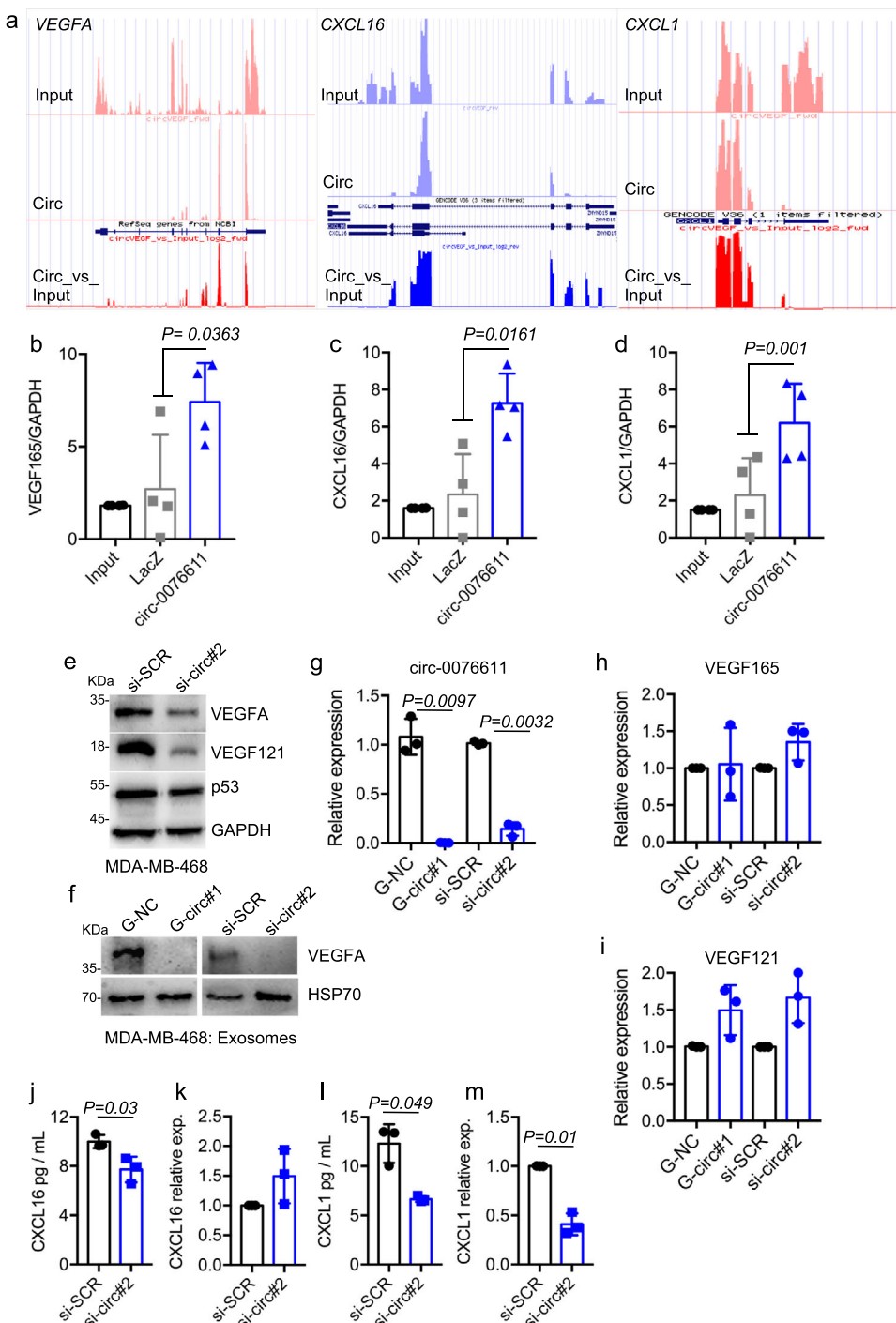

MDA-MB-468 cells (Fig. 5a). Similar results were obtained in non-transformed MCF10A mammary epithelial cells (showing no detectable basal circ_0076611 in ISH) after transfection with a circ_0076611 expression vector (Fig. 5b). Consistently, cell fractionation evidenced both nuclear and cytoplasmic expression of circ_0076611 (Fig. 5c), while expression of MALAT1 lncRNA, used as control, was detected specifically in the nucleus. The results of the localization analysis prompted us to evaluate whether circ_0076611 is involved in translation control. ChIRP assays evidenced the ability of circ_0076611 to interact with 5.8S, 18S and 28S ribosomal RNA (rRNA) and with nucleolar protein fibrillarin (FBL) (Fig. 5d and Supplementary Fig. 3a), being this last also confirmed by RIP assays (Supplementary Fig. 3b). We next evaluated rRNA profiles through bioanalyzer electropherogram

analysis in MDA-MB-468 cells depleted or not of circ_0076611. Quantification of rRNA peak areas showed a decrease of both 18S and 28S forms upon silencing of circ_0076611 (Fig. 5e, f and Supplementary Fig. 3c). On this basis, we next sought to investigate whether circ_0076611 depletion impacts on translation rate. We first measured the levels of newly synthesized polypeptides using a puromycin incorporation assay in cells depleted or not of circ_0076611. As shown in Fig. 5g, silencing of circ_0076611 caused a decrease of translation rate in MDA-MB-468 cells. Consistently, polyribosome (polysome) fractionation analysis evidenced a drop of polysome fraction in samples depleted of circ_0076611 (Fig. 5h, i). Analysis of circ_0076611 in the polyribosomal fractions evidenced that this circRNA mainly localizes in the 40/60S fractions (Fig. 5j), as well as in the fraction containing

**Fig. 4 Circ_0076611 controls the expression of VEGFA protein. a** Representative tracks from ChIRP-RNAseq results for VEGFA, CXCL16 and CXCL1 genes in MDA-MB-468 cells. Tracks from input and circ_0076611 (Circ) samples, along with the circ_0076611/input ratio value (Circ_vs_input) are presented. RT-qPCR analysis of VEGF165 (**b**), CXCL16 (**c**) and CXCL1 (**d**) mRNAs in circ_0076611 ChIRP assays performed in MDA-MB-468 cells to validate ChIRP-RNAseq results. A biotinylated oligonucleotide complementary to circ_0076611 and a LacZ oligonucleotide (as negative control) were used in ChIRP assays. Results ($N \geq 3$ independent biological replicates) were normalized to GAPDH mRNA level and expressed as log2. Efficiency of circ_0076611-ChIRP is shown in Fig. 3a. (*p* value has been calculated by paired, two-tailed, Student's *t* test). **e** Western blot analysis of the indicated proteins in MDA-MB-468 cells transfected with a siRNA to silence circ_0076611 (si-circ#2) or control siRNA oligo (si-SCR) for 72 h. **f** Western blot analysis of the indicated proteins in protein extracts from exosomes isolated from the conditioned medium of MDA-MB-468 cells transfected with GapmeRs or siRNAs to silence circ_0076611 (G-circ#1 and si-circ#2, respectively) or control oligos (G-NC, si-SCR) for 72 h. HSP70 protein has been evaluated as exosome marker. RT-qPCR analysis of circ_0076611 (**g**), VEGF165 (**h**) and VEGF121 (**i**) isoforms in MDA-MB-468 cells transfected with GapmeRs or siRNAs to silence circ_0076611 (G-circ#1 and si-circ#2, respectively) or control oligos (G-NC, si-SCR) for 72 h. *p* value has been calculated by paired, two-tailed, Student's *t* test on $N = 3$ independent biological replicates. Expression of CXCL16 and CXCL1 proteins and mRNAs, evaluated, respectively, by ELISA assay (**j-l**) and RT-qPCR (**k-m**), in MDA-MB-468 cells transfected with a siRNA to silence circ_0076611 (si-circ#2) or control siRNA oligo (si-SCR) for 72 h. ELISA assay has been performed on conditioned media from MDA-MB-468 cells. *p* value has been calculated by paired, two-tailed, Student's *t* test on $N = 3$ independent biological replicates. Bars indicate standard deviation.

free RNAs, which suggested a possible involvement in translation initiation. To address this possibility, we assessed the interaction between circ_0076611 and proteins of the eIF4F cap-binding complex, involved in the first step of translation initiation[36]. We first tested EIF4B, a component of eIf4F necessary for the RNA unwinding activity of the complex[36]. Specifically, we verified by RIP assay that circ_0076611 actually interacts with EIF4B (Fig. 5k). Interaction of circ_0076611 with EIF4G, an additional component of the eIF4F complex able to interact with poly(A) binding protein and to circularize the RNA during translation[36], further supported the involvement of this circRNA in translation initiation (Supplementary Fig. 3d). Of note, circ_0076611 silencing impaired the interaction between EIF4B protein and the 5'-UTR regions of circ_0076611 target mRNAs, such as CCNB2 and VEGFA (Fig. 5k and Supplementary Fig. 3e). No impact of circ_0076611 on the overall EIF4B protein levels was observed (Supplementary Fig. 3f). Taken together, these results suggest that circ_0076611 may improve the initiation of translation by promoting the interaction between components of the eIF4F complex and its interacting mRNAs in TNBC cells.

**lncRNA MALAT1 and ID4 protein control circ_0076611 expression in TNBC cells.** We next explored the mechanisms at the basis of circ_0076611 expression in TNBC cells. As already mentioned, VEGFA isoforms abundance is regulated, in BC cells, by a RNP complex composed of lncRNA MALAT1 and protein factors SRSF1, ID4 and mutant p53[12]. RT-PCR analysis of circ_0076611 in MDA-MB-468 cells (carrying endogenous hot-spot mutation p53R273H) showed reduced expression in cells silenced for MALAT1, ID4 and mutant p53, compared to control (Fig. 6a and Supplementary Fig. 3g), along with increased expression of the longer isoforms containing Exon 7, as expected from our previous study[12]. Sanger sequencing verified the identities of PCR products. Decreased expression of circ_0076611 following single MALAT1 silencing was observed by RT-qPCR using multiple siRNAs or an ASO in multiple TNBC cell lines (Fig. 6b). Decreased expression of circ_0076611 was confirmed also upon depletion of ID4 protein by siRNA transfection (si-ID4, Fig. 6c–f) or by CRISPR/Cas9-mediated gene editing (ID4-KO, Fig. 6d–f). Consistently, MDA-MB-468 ID4-KO cells also showed reduced protein expression of circ_0076611 targets c-MYC and BRD4 (Fig. 6g).

It is well recognized that the biogenesis of a circRNA, containing one or more exons, depends on: (1) the presence of inverted-repeated sequences (favoring base-pairing) in flanking introns; (2) the presence of consensus sequences for specific RBPs (RNA-binding proteins (RBPs)) which may enhance or inhibit back-splicing[35]. As inverted-repeated sequences are present in the

intronic regions flanking VEGFA exon 7 (Fig. 7a and Supplementary Fig. 4a, b), we cloned this DNA fragment in a pCDNA3.1 expression vector, which showed high expression efficiency of circ_0076611 when transfected in BC cells (Fig. 7b). A vector lacking the downstream repeated sequence in intron 7 (CIRC#del, Fig. 7a) showed no circ_0076611 overexpression (Fig. 7b), further indicating the requirement of these intronic fragments for efficient circ_0076611 production. Of note, efficiency of circ_0076611 overexpression was significantly reduced in cells depleted of MALAT1 (Supplementary Fig. 4c), further indicating the requirement of MALAT1 for circ_0076611 expression.

Bioinformatic prediction and mapping of binding sites for RBPs (http://rbpmap.technion.ac.il/, http://pridb.gdcb.iastate.edu/RPISeq, https://circinteractome.nia.nih.gov/) in intronic regions flanking exon 7 highlighted candidate RBPs (SRSF1, hnRNPA1, EIF4A3, QKI and PTBP1), possibly affecting circ_0076611 expression, of which only PTBP1 and SRSF1 were confirmed to interact with circ_0076611 pre-mRNA in RIP experiments (Fig. 7c and Supplementary Fig. 5). Silencing of PTBP1 or SRSF1 by siRNA transfection, however, revealed opposite functions of these RBPs, with PTBP1 inhibiting and SRSF1 favoring the expression of circ_0076611 (Fig. 7d, e). On this basis, we hypothesized that the RNP complex comprising lncRNA MALAT1 and SRSF1, ID4 and mutant p53 proteins could enhance the expression of circ_0076611 by inhibiting PTBP1 activity while favoring that of SRSF1 (Fig. 7f). To address this, we first evaluated whether or not the whole RNP complex interacts with PTBP1. By RIP experiments, we observed that PTBP1 protein indeed interacts with MALAT1 (Fig. 7g) and with the protein members of the complex, namely ID4 (Fig. 7h), mutant p53 and SRSF1 (Fig. 7h and Supplementary Fig. 4d) in MDA-MB-468 and HCC1395, TNBC cell lines carrying the two hotspot mutants p53R273H and p53R175H, respectively. In support of our hypothesis we observed by RIP experiments a reduction of SRSF1 (Fig. 7i) and an increase of PTBP1 (Fig. 7j) protein recruitment on circ_0076611 pre-mRNA in MDA-MB-468 ID4-KO cells, compared to control. Finally, we assessed that PTBP1 silencing is able to rescue the inhibitory effect of MALAT1 depletion on circ_0076611 expression, further supporting a role for MALAT1 in PTBP1 protein inhibition (Fig. 7k, l).

**ID4 protein associates with circ_0076611 expression in breast cancer.** In light of the involvement of ID4 protein in the control of circ_0076611 expression, we evaluated whether these two factors are significantly associated in BC. High ID4 expression is indeed associated to TNBC and, specifically, to those TNBC with basal phenotype, namely basal-like breast cancer (BLBC), where

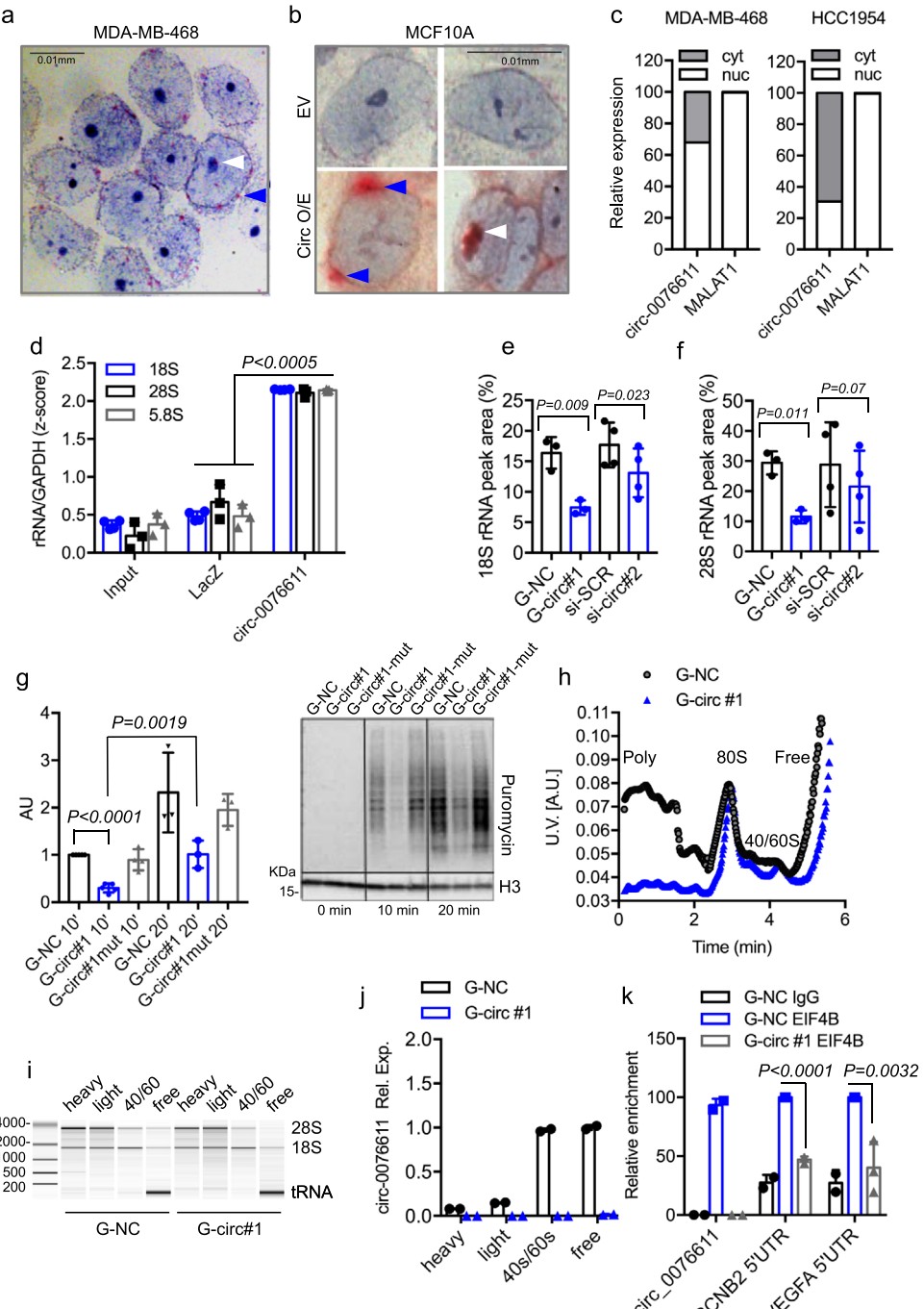

ID4 marks a subset of poor prognosis cancers[12,24,26,37]. We performed immunohistochemistry (IHC) for ID4 on the same TMA used for circ_0076611 ISH. This allowed highlighting a higher percentage of cases positive for circ_0076611 in the ID4-high and ID4-medium populations compared to the ID4-low population in BLBC (Fig. 8a, b).

To evaluate the impact of the ID4/circ_0076611 axis on BLBC prognosis we used the list of circ_0076611 target mRNAs as surrogate for the expression of circ_0076611. Specifically, we first intersected the list of circ_0076611 targets with mRNAs negatively or positively correlated to ID4 mRNA in the TCGA dataset of BLBC (Fig. 8c)[38], obtaining, respectively, two signatures of 19 and 27 genes that are ID4-correlated and circ_0076611-associated. Evaluation of the clinical relevance of these signatures by Kaplan–Meier analysis in BLBC showed that they significantly

predict distant-metastasis-free survival (DMFS) and recurrence-free survival (RFS) both when combined (Fig. 8d) and when tested individually (Fig. 8e). Specifically, low expression of the 19 negatively-correlated (19dn) and high expression of the 27 positively-correlated (27up) mRNAs predicted a worse prognosis (Fig. 8d, e). Collectively, these results indicate that circ_0076611 is a target of ID4 protein in BLBC and that the ID4/circ_0076611 axis controls a gene expression program relevant for the prognosis of this BC subtype.

## Discussion

Our findings show that lncRNA MALAT1 not only favors the expression of those splicing isoforms of VEGFA lacking exon 7, as we have already reported[12], but also orchestrates a back-splicing event from exon 7 which leads to the formation of a

**Fig. 5 Circ_0076611 impacts on the translation rate of its target mRNAs.** Analysis by in situ hybridization of circ_0076611, performed by using the BaseScope technology with a probe complementary to the circRNA back-splice junction, in MDA-MB-468 cells (**a**) and in MCF10A cells (**b**) transfected with an empty pCDNA3.1 vector (EV) or a circ_0076611 expression vector (circ O/E). Blue arrows indicate perinuclear staining while white arrows indicate nucleolar staining. Scale bar 0.01 mm. **c** Distribution of circ_0076611 and MALAT1 RNAs in nucleus and cytoplasm of HCC1954 and MDA-MB-468 cells. Expression has been evaluated by RT-qPCR on RNA preparations obtained by cell fractionation. **d** Evaluation by RT-qPCR of ribosomal RNAs 18S, 28S and 5.8S in circ_0076611-ChIRP assays. A biotinylated oligonucleotide complementary to circ_0076611 and a LacZ oligonucleotide (as negative control) were used in ChIRP assays. Results were normalized to GAPDH mRNA level and expressed as z-scores. Efficiency of circ_0076611-ChIRP is shown in Fig. 3a. p value has been calculated by paired, two-tailed Student's t test on N = 3 independent biological replicates. **e**, **f** 18S and 28S rRNA peak areas from Bioanalyzer electropherogram analysis of total RNA (using Agilent RNA 6000 Nano Kit) from MDA-MB-468 cells transfected with GapmeRs or siRNAs to silence circ_0076611 (G-circ#1 and si-circ#2, respectively) or control oligonucleotides (G-NC, si-SCR) for 72 h. p value has been calculated by ratio paired, two-tailed Student's t test (N ≥ 3 independent biological replicates). **g** Puromycin incorporation for the indicated time points, detected by western blot analysis in MDA-MB-468 cells depleted of circ_0076611 (si-circ) and in control cells (si-NC, si-circ-mut). Cells were silenced for circ_0076611 for 48 h and then challenged with 10 μg/ml of puromycin (#ant-pr-1, InvivoGen), a tyrosyl-tRNA mimic that blocks translation, for 10 or 20 min. The level of newly synthesized polypeptides was evaluated by using an anti-puromycin antibody (#MABE343, Millipore) in western blot. Quantification of puromycin signal normalized to histone H3 protein is shown in the graph while a representative western blot is shown in the right panel. p value has been calculated by paired, two-tailed Student's t test (N ≥ 3 independent biological replicates). **h**, **i** Representative polysome profiles obtained by sucrose gradient fractionation in cytoplasmic extracts from MDA-MB-468 cells silenced for circ_0076611 expression (G-circ#1) and in control cells (G-NC) (**h**). A Bioanalyzer electropherogram analysis (**i**) has been carried out using Agilent RNA 6000 Nano Kit, loading equal amounts of RNA extracted after sucrose gradient fractionation from the indicated pools of fractions (heavy, light, 40/60S and free RNAs) to control RNA quality. **j** RT-qPCR analysis of circ_0076611 to evaluate its distribution in the indicated pools of fractions (heavy, light, 40S/60S and free RNAs) obtained by sucrose gradient fractionation as in **h**, **i**. **k** RIP assay to evaluate the enrichment of circ_0076611, CCNB2 and VEGFA mRNAs in samples immunoprecipitated with anti-EIF4B antibody or IgG in control cells (G-NC) or in circ_0076611-depleted (G-circ#1) MDA-MB-468 cells crosslinked with formaldehyde. p value has been calculated by paired, two-tailed Student's t test (N = 3 independent biological replicates). Bars indicate standard deviation.

circular RNA, circ_0076611, previously identified in the study by ref. [39] and reported in the circBase database (http://www.circbase.org). Alteration of a group of 50 circRNA following MALAT1 depletion had been previously reported in Jurkat cells[40] but, to our knowledge, no other studies are present related to the ability of MALAT1 to control circular RNAs expression in cancer.

We observed that ID4 protein also participates in this function of MALAT1. Accordingly, a significant association between ID4 protein expression and the presence of circ_0076611 has been evidenced by the analysis of BLBC cases. As ID4 is a predictor of survival in BLBC, it will be important in the future to define whether circ_0076611 expression is also relevant for BLBC survival. At the moment, we could only verify that some signatures of transcripts interacting with circ_0076611 in TNBC cells and related to ID4 are actually associated with survival (DMFS and RFS) in BLBC. The requirement of ID4 protein for circ_0076611 expression could also explain why circ_0076611 is not ubiquitously expressed in the TNBCs analyzed. ID4 protein indeed has been reported as expressed in 70% of cases, with only 30% of TNBC showing high ID4 score[14,41].

ID4 protein had been previously shown as required by MALAT1 to exert its regulatory role on VEGFA splicing[12]. Of note, ID4 protein is required in BC cells for efficient interaction between MALAT1 and splicing factor SRSF1[12], being this last responsible for VEGFA pre-mRNA splicing control[12,42,43]. The present study highlights that SRSF1 is also able to regulate the back-splicing event that occurs on VEGFA exon 7. Silencing of SRSF1 indeed leads to circ_0076611 downregulation; however, as this could depend on the reduction of VEGF121 isoform (characterized by exon 7 exclusion) that is observed in the absence of SRSF1[12], we evaluated if SRSF1 could directly control exon 7. Two predicted binding sites for SRSF1 are present in the intronic regions flanking VEGFA exon 7 and we observed that SRSF1 protein actually binds to VEGFA pre-mRNA exon 7/intron 7 boundary in RIP experiments (Fig. 7a). These results strongly support a direct role of SRSF1 in the control of exon 7 back splicing.

Besides the positive role of SRSF1 in the promotion of circ_0076611 expression, we also evidenced that the whole RNP complex containing SRSF1 is able to bind to a negative regulator of circ_0076611 expression, the Polypyrimidine tract binding protein 1 (PTBP1) protein. It is highly reasonable that the RNP complex counteracts the activity of PTBP1. Contrarily to our results about circ_0076611 expression, PTBP1 had been previously shown to induce the expression of another circRNA, circRNA_001160, in glioma endothelial cells[44].

We found that circ_0076611 interacts with 321 mRNAs in MDA-MB-468 cells and possibly controls their expression. We do not know if the interaction between circ_0076611 and its targets is direct or, otherwise, mediated by some RBP. Among the targets of circ_0076611, we identified VEGFA mRNA. VEGFA post-transcriptional regulation is exerted through numerous mechanisms in cancer cells, such as for example the HuR- and NF90-dependent mRNA stabilization or microRNA-dependent regulation of translation[45]. Here, we identified that circ_0076611 regulates VEGFA translation through the enhancement of the interaction between translation initiation machinery and VEGFA transcript. Specifically, circ_0076611 is recruited on the exon 8A region of VEGFA mRNA, a region enclosed in all linear VEGF$_{XXX}$ isoforms, suggesting that circ_0076611 enhances the protein expression of the whole set of VEGF$_{XXX}$ transcripts present in the cell. Binding between circ_0076611 could be likely mediated by some of the many so far identified VEGFA regulators. The exon 8A region, in particular, is targeted by SRSF1 protein, involved in the selection of the proximal splice site. Others and we have previously shown that SRSF1 interacts with pre-VEGFA as well as with mature VEGFA mRNA[12]. Moreover, SRSF1, continuously shuttling between the nucleus and the cytoplasm, associates with translating ribosomes and stimulates translation[46,47]. We cannot therefore exclude an involvement of SRSF1 in the functional activity of circ_0076611 on VEGFA exon 8.

Another circular RNA has already been identified, circSMARCA5, which has an impact on the expression of VEGFA[48]; however, differently from circ_0076611 which enhances VEGFA translation, circSMARCA5 affects VEGFA splicing, favoring the pro- to anti-angiogenic VEGFA isoforms ratio, thanks to its ability to sponge SRSF1 protein in glioblastoma multiforme cells.

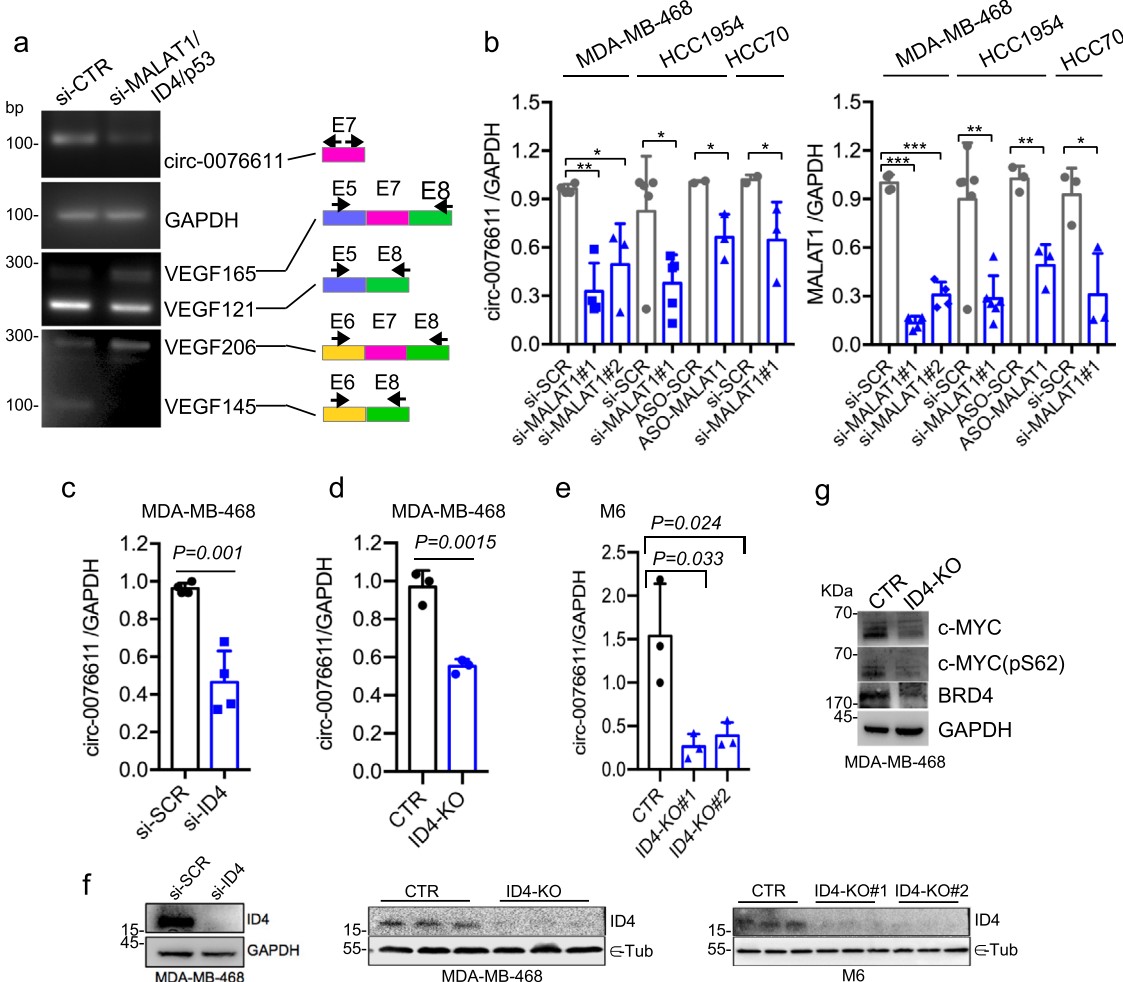

**Fig. 6 MALAT1 and ID4 induce circ_0076611 expression in TNBC cells. a** RT-PCR analysis of circ_0076611 and of VEGFA linear transcripts in control (si-CTR) MDA-MB-468 cells and in cells depleted of ID4/MALAT1/mutant p53 expression, by transfection of siRNAs to ID4 and p53, as well as of ASO to MALAT1, has been performed using the primers indicated on the right. The identity of PCR products has been verified by Sanger sequencing and is indicated on the right. Sequences of siRNA and ASO are enclosed in Supplementary Table 1. **b** Expression of circ_0076611 (left) and MALAT1 (right), evaluated by RT-qPCR in the indicated breast cancer cell lines transfected with siRNAs or ASOs directed to MALAT1 (si-MALAT1#1, si-MALAT1#2, ASO-MALAT1) or controls (si-SCR, ASO-SCR). **c–e** Expression of circ_0076611 in MDA-MB-468 and M6 breast cancer cells silenced for ID4, by siRNA transfection (**d**) or by CRISPR/Cas9-mediated gene editing (**d**, **e**). **f** Western blot of ID4 protein in the experimental conditions analyzed in **c–e**. **g** Western blot analysis of the indicated circ_0076611 targets in control (CTR) MDA-MB-468 and in CRISPR/Cas9-mediated ID4-KO cells. Data are presented as mean plus standard deviation. *$p \leq 0.05$, **$p \leq 0.005$, **$p \leq 0.0o05$ (paired, two-tailed Student's *t* test). Results from at least three biological replicates are shown.

The functional output of increased expression of VEGFA and of additional angiogenesis regulators, such as CXCL1 (GRO-alpha) and CXCL16, driven by circ_0076611, is likely increased angiogenesis and increased proliferation potential of TNBC cells, resulting from an autocrine signaling[49,50]. Interestingly, CXCL1 had been also previously reported as a post-transcriptional target of ID4 protein in BC cells[41]. The observation that circ_0076611 is expressed at a higher level in mammosphere cultures than in control adherent cells also suggests its possible involvement in stemness, a role already reported for ID4[51].

An extremely interesting feature of many circRNA is their release by tumor cells in the extracellular space, enclosed in vesicles such as exosomes, enabling delivery to surrounding cells in the tumor microenvironment and transport through the bloodstream[35]. The amount of circRNA secretion in exosomes was also related to the expression of the microRNAs bound to them[35]. We here detected circ_0076611 both in exosomes released by cultured tumor cells and in the serum of BC patients.

Functionally, CM from circ_0076611-overexpressing cells is able to increase the proliferative and colony-forming potential of TNBC cells, suggesting that the release of circ_0076611 might amplify the mitogenic signal in TNBC. The significance of the presence of circ_0076611 in the blood is not yet known and deserves to be investigated in further studies. However, the observation that BC patients with elevated serum levels of circ_0076611 show a decrease in this circRNA after treatment with metformin, despite being only a preliminary finding, suggests that this circRNA may represent a biomarker in BC. In recent years, we have seen a growing interest in the possible use of anti-diabetic agents, such as metformin, in the management of BC. Preclinical research provided intriguing insight into the cellular mechanisms behind the oncostatic effects of Metformin, even in TNBC[52]. Specifically, Metformin reduces the percentage of TNBC stem cells through glycogen synthase kinase-3β-dependent downregulation of KLF5, a crucial stem cell transcription factor in basal-type TNBC cells, which promotes TNBC

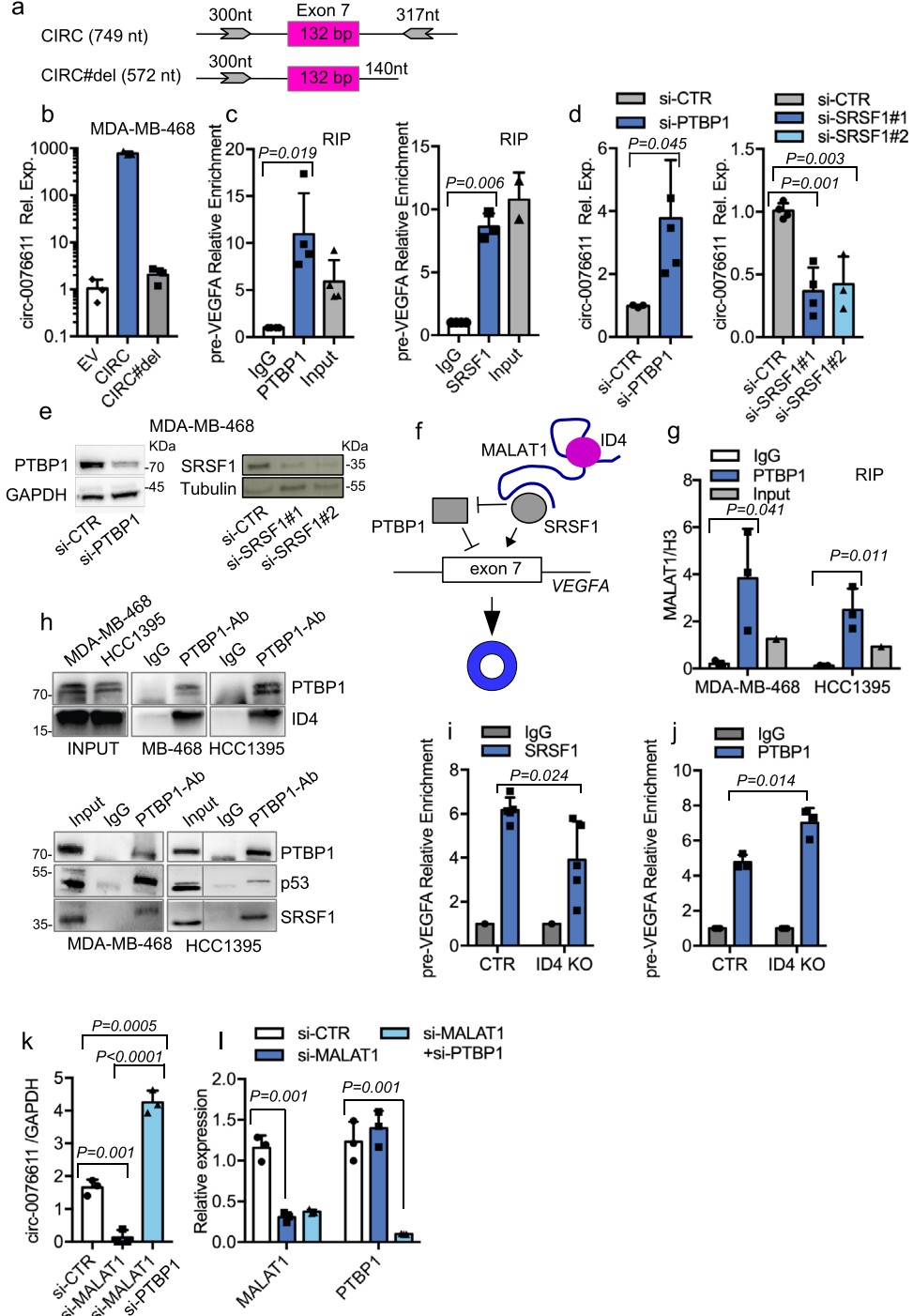

cell proliferation, survival, migration, invasiveness and stemness. The possible ability of metformin to modulate the expression level of circ_0076611 in BC cell lines and the molecular mechanisms underlying this regulation in TNBC are currently under study.

## Materials and methods

**Cell cultures and transfections.** All cell lines were grown at 37 °C, 5% $CO_2$. SK-BR-3 (ATCC® HTB-30™), HCC70 (ATCC® CRL2315™), HCC1954 (ATCC® CRL2338™), MDA-MB-231 (ATCC® HTB-26), HCC1395 (ATCC® SC-CRL-2324) and HCC1143 (ATCC® CRL-2321) cell lines were maintained in RPMI medium containing 10% heat-inactivated fetal bovine serum (FBS, Gibco) and penicillin/streptomycin. MCF10A (ATCC® CRL-10317) cells were maintained in DMEM F12 (Gibco) supplemented with 5% horse serum plus 0.01 mg/ml insulin, 0.5 ug/ml hydrocortisone and 20 ng/ml EGF. MDA-MB-468 (ATCC® HTB-132) were

maintained in DMEM high glucose (Euroclone) containing 10% FBS, L-Glutamine and penicillin/streptomycin. Ovcar-3 (ATCC® HTB-161) were maintained in RPMI medium containing 20% heat-inactivated FBS (Gibco), 0.01 mg/ml insulin and penicillin/streptomycin. SUM149PT and SUM159PT cells were maintained in RPMI medium containing 10% heat-inactivated FBS (Gibco), 0.01 mg/ml insulin and penicillin/streptomycin. Cell lines were periodically tested for *Mycoplasma* contamination by PCR analysis. For transfections, siRNAs (synthesized by IdT or Eurofins) (300 pmol) and LNA™ antisense GapmeRs (synthesized by Qiagen) (30 pmol), were transfected with RNAiMax reagent (Thermo Fisher Scientific, Waltham, MA, USA) using $3 \times 10^5$ cells in 2.5 ml of medium for 48 h, unless otherwise stated, following manufacturer's instructions. Supplementary Table 1 includes the sequences of siRNAs and LNA™ antisense GapmeRs oligonucleotides used for transfections.

CM from MDA-MB-468 cells, overexpressing circ_0076611 or control cells, were prepared by culturing equal cell numbers for each condition for 24 h in serum-free medium. CM were centrifuged to eliminate cell residues, at $3000 \times g$ for 15 min, before preparation of aliquots and storage at −80 °C.

**Fig. 7 MALAT1 and ID4 favor circ_0076611 expression by counteracting RNA-binding protein PTBP1. a** Schematic representation of the genomic regions cloned in pCDNA3.1 vector to generate a circ_0076611 expression vector (indicated as CIRC) and a deletion mutant lacking an inverted repeated sequence required for circ_0076611 expression (indicated as CIRC#del). **b** Expression of circ_0076611 ($\log_{10}$), evaluated by RT-qPCR, in MDA-MB-468 cells transiently transfected with a pCDNA3.1 vector (EV) or with the circ_0076611 expression vectors indicated in **a**. **c** Ribonucleoprotein Immunoprecipitation assays (RIP) in MDA-MB-468 cells (after UV crosslinking) showing the enrichment of VEGFA precursor RNA (pre-VEGFA) in samples immunoprecipitated with control IgG or with antibodies directed to PTBP1 (left) or SRSF1 (right) proteins. Data are normalized to 28S rRNA and presented as folds of enrichment over IgG. **d, e** Expression of circ_0076611, evaluated by RT-qPCR, in MDA-MB-468 cells transfected with a pool of siRNAs directed to PTBP1 (left) or two different siRNAs directed to SRSF1 for 72 h (right) (**d**). WB showing the efficiency of PTBP1 and SRSF1 interference after 72 h of siRNA transfection is shown in **e**. **f** Scheme summarizing the regulatory roles of lncRNA MALAT1 and of proteins ID4, SRSF1 and PTBP1 in the control of circ_0076611 back-splicing. **g, h** RIP assay in the indicated cell lines (after UV crosslinking) showing the level of MALAT1 lncRNA, evaluated by RT-qPCR, in samples immunoprecipitated with control IgG or with an antibody directed to PTBP1 protein (**g**). A fraction (25%) of eluates recovered during RIP experiments has been used for protein analysis by WB to control PTBP1 IP and co-IP of ID4, mutant p53 and SRSF1 proteins (**h**). Enrichment of SRSF1 (**i**) and PTBP1 (**j**) proteins on pre-VEGFA mRNA evaluated by RIP assay in control MDA-MB-468 cells (CTR) and in ID4 knocked-out counterpart (ID4-KO) (after formaldehyde crosslinking) by using antibodies directed to SRSF1 or PTBP1 and IgG as negative control. Data are presented as folds of enrichment over IgG. Expression of circ_0076611 (**k**) and MALAT1 (**l**) and PTBP1 (**l**), evaluated by RT-qPCR, in MDA-MB-468 cells transfected with siRNAs directed to MALAT1, or to MALAT1 and PTBP1, for 72 h. Data are presented as mean plus standard deviation. *p* values have been calculated by paired, two-tailed Student's *t* test. Results from at least three biological replicates are shown. Bars indicate standard deviation.

**Mammosphere formation assay**. Mammospheres were generated by placing HCC70 and HCC1954 cells in suspension ($10^3$ cells/ml) in mammosphere media containing DMEM/F12 supplemented with 20 ng/ml EGF, 0.01 ml/ml insulin (Humulin R), B27 (1:50; Invitrogen), 0.4% BSA, 100 U/ml penicillin, 100 mg/ml streptomycin. Experiments were performed using ultra-low attachment plates (Corning). Spheres were grown at 37 °C, 5% $CO_2$ for 6–8 days, without disturbing the plates.

**Cell viability and colony-formation assays**. MDA-MB-468 and HCC1954 cells were transfected, as indicated above, with GapmeR oligonucleotides. Viability of transfected cells was assessed using ATPlite assay (Perkin Elmer, Massachusetts, USA) at the indicated time points, according to the manufacturer's instructions. Cells ($8 \times 10^2$ cells) were seeded in 96-well plates and cultured for 24–48–72 h. Each plate was evaluated immediately on a microplate reader (Expire Technology, Perkin Elmer). Calcusyn software was used to calculate combination index. For colony-formation assays, $3 \times 10^5$ MDA-MB-468 cells were transfected with Gap-meR oligonucleotides, as described above, plated in 6-well multi-well plates and 24 h post-transfection trypsinized and plated in 6-well plates (400 cells per well). Stable clones of MDA-MB-468 overexpressing circ_0076611 or knocked down for ID4 were directly plated in 6-well plates (400 cells per well). Fresh medium was added twice a week. Colonies were fixed and colored with crystal violet after 3 weeks of culture. Counting was performed manually.

**Wound-healing assay**. To study the coordinated movement of a cell population, a wound-healing assay was used. Cells were plated in a 24-well cell culture plate to form confluent monolayers. After 24 h of culture, a pipette tip was used to create a scratch in each well. After the scratch, cell culture was rinsed twice with phosphate buffered saline (PBS) 1x, than medium was replaced with serum-free medium. A horizontal reference line on the bottom of each well was made to have a grid for alignment and obtain the same field for each image acquisition. The images of the gap were acquired at 0 h (immediately after scratching) and after 24 h of incubation at 37 °C with 5% $CO_2$. The scratch area was quantified using the open source imageJ/Fiji®, calculating the percentage of wound closure.

**Invasion assay**. MDA-MB-231 cells were transfected with LNA™ antisense GapmeR oligonucleotides, as described. After 24 h, permeable inserts (8.0 μm pore diameter) were coated with Corning® Matrigel® matrix. In total, $6 \times 10^4$ cells were plated in the upper insert in 500 μl of serum-free DMEM, while 750 μl of DMEM 10% FBS were added in the lower well. The invasion assay was carried out for 24 h at 37 °C with 5% $CO_2$. Transwell inserts were washed twice with PBS 1x; the cells on the inside of the insert were gently removed with a cotton swab, while the cells on the lower side were stained with crystal violet for 15 min, and then washed twice with PBS 1x. The images were acquired under a microscope. For quantification, bound crystal violet was eluted with 33% acetic acid. The eluate was transferred to a 96-well plate and the absorbance at 590 nm was measured, using Multiskan GO UV/Vis microplate spectrophotometer (Thermo Scientific). A standard curve was generated, in parallel, to allow quantification by plating cells at different densities.

**Breast cancer samples**. A commercially available TMA has been used to analyze circ_0076611 expression by ISH and ID4 protein expression by IHC. Specifically, we have used the TMA #BR1301 from US Biomax, Inc. Analysis of circ_0076611 expression by RT-qPCR has been carried out on 12 tumor and peritumor tissues from ER-negative BC. Collection of tumors from BC patients was reviewed and approved by the ethics committee of the Regina Elena National Cancer Institute (IFO1270/19) and contained data for which written informed consent was obtained from all patients.

**Immunohistochemistry (IHC)**. The immunohistochemical assessment of 127 cases of triple-negative BC on TMA was performed with the following antibodies: ID4 (clone: B-5) mouse monoclonal antibody (Santa Cruz), overnight incubation at the dilution of 1:100 (pH6); CK5 (clone: XM26) mouse monoclonal antibody (Novocastra™) at the dilution of 1:100 (pH6). Immunoreactions were revealed by Bond Polymer Refine Detection System on an automated auto-stainer (Bond III™ Stainer, Leica Biosystem, Milan, Italy). Whole slide images of ID4 and CK5 IHC were acquired by scanning of the original glass slides using Aperio CS2 whole slide scanner (Leica Biosystems, Germany). The default autofocus mode was used, but in a few cases manual focus was applied. The image files (.svs format) were stored on a Network Attached Storage running with the Aperio ScanScope software. Staining intensity for ID4 was evaluated as: 0 negative, 1+ weak, 2+ moderate, 3+ strong. ID4 was considered negative when less than ≤5% of the neoplastic cells exhibited nuclear immunoreaction. Staining intensity for CK5 was evaluated as: 0 negative, 1+ weak, 2+ moderate, 3+ strong as a percentage of positive neoplastic cells.

**Treatments with RNase R/RNase H**. To detect circ_0076611, total RNA was initially treated with DNase (DNAfree, Ambion) and, subsequently, with RNase R, an RNA exonuclease that can degrade linear RNA and keep the circRNA intact, and/or RNase H, an RNA exonuclease that can degrade RNA/DNA hybrids. Briefly, 1U RNaseR (Lucigen) was used to treat 1 μg of total RNA for 1 h and 40 min at 37 °C; RNaseR was inactivated at 65 °C for 20 min. To perform RNaseH treatment, RNA was first incubated with an exon 7 ASO (10 pmol/1 μg RNA) of sequence 5'-AGG AAC ATT TAC ACG TCT GCG-3' for 10 min at 65 °C followed by slow cooling to 37 °C, to allow hybridization; then RNaseH (E. Coli) was added (1U/1 μg RNA) and incubated 40 min at 37 °C. The abundance of circ_0076611, VEGF165 and GAPDH transcripts was evaluated by RT-qPCR.

**RNA isolation and qPCR**. Total RNA was isolated with TRIzol reagent (Thermo Fisher Scientific) and its concentration was measured using a NanoDrop 2000 (Nanodrop Technologies, Wilmington, DE, USA). The integrity of RNA samples (RIN) and quantification of ribosomal RNA peaks (18S and 28S) were assessed using Agilent 2100 Bioanalyzer (RNA 600 nano kit, Agilent Technologies). In total, 2 μg of total RNA was used in a reverse transcription reaction followed by qPCR analysis. Reverse transcription was performed with SuperScript™ IV VILO™ Master Mix (Invitrogen), following manufacturer's instructions but performing the reverse transcription reaction at 65 °C. PCR was carried out on ABI 2020 Instrument. qPCR was carried out on QuantStudio5 Fast Sequence Detection Systems (Applied Biosystems, Carlsbad, CA, USA). Primers used for PCR and qPCR analyses are listed in Supplementary Data 1. The expression values of mRNAs were calculated by ΔΔCt method and normalized to GAPDH or on the indicated control genes.

**Analysis of circular RNA expression**. In total, 1 μl of cDNA was used in PCR to analyze circRNA expression in PCR and in real-time PCR. PCR assays were carried out in a 20 μl reaction using AmpliTaq Gold™ 360 Master Mix (Thermo Fisher Scientific, Cat. No. 4398886) following manufacturer's instructions using 10 pmol of each primer. PCR was carried out on ABI 2020 Instrument (Applied Biosystems, Carlsbad, CA, USA). In total, 58 °C/30 s was used as annealing condition and 72 °C/30 s as extension condition. Primers used in PCR for circ_0076611 are enclosed in Supplementary Data 1 (indicated as div_F and div_R2). PCR products were evaluated after 38 cycles of PCR on a 3% (w/v) agarose gel. For real-time PCR analysis of circ_0076611, primers div_F2/div_R2 and a TaqMan probe covering the 5'-3' junction (see Supplementary Data 1) were used. In total, 15 μl reactions were set up using the Luna® Universal qPCR Master Mix (New England Biolabs, Cat. No. BM3003E); 10 pmol of each primer and 5 pmol of TaqMan probe. qPCR was

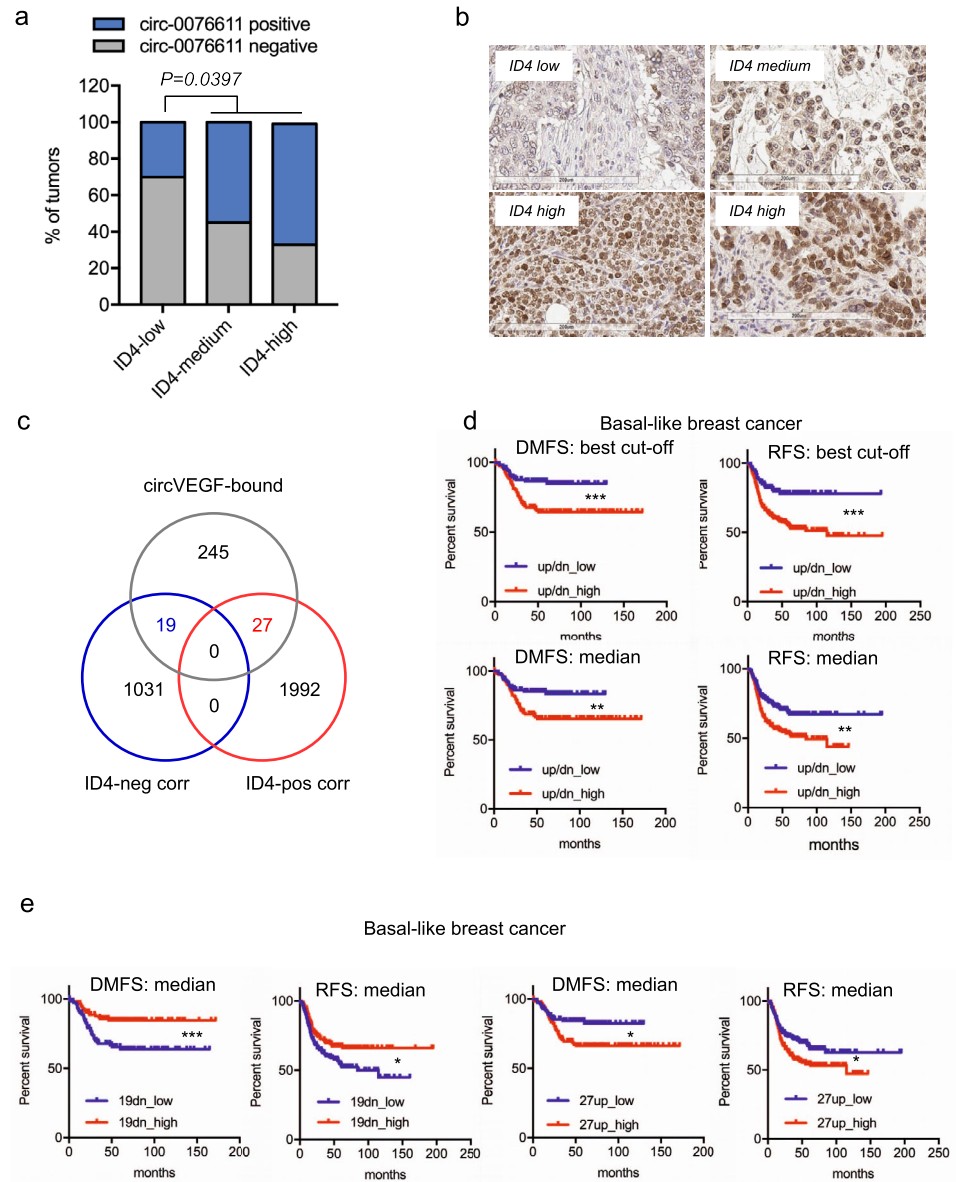

**Fig. 8 Circ_0076611 and ID4 protein expression are associated in triple-negative breast cancer. a, b** Distribution of circ_0076611 expression in tumors from the TMA (Ck5+) characterized by different strength of ID4 protein expression. Two-tailed $p$ value has been calculated by Chi-square test comparing circ_0076611-negative vs. circ_0076611-positive cases, in the two groups with ID4-low vs. ID4-medium/high. Representative images of various ID4 staining intensities, as assessed by immunohistochemistry (IHC), are shown in **b**. **c** Venn diagram indicating the intersection between circ_0076611 target mRNAs and transcripts positively or negatively correlated to ID4 in the TCGA basal-like breast cancer dataset. 19-negatively and 27-positively ID4-correlated genes that are also bound by circ_0076611 were identified. **d** Kaplan–Meier analysis showing the association between the expression of the two combined signatures of 19-negatively (19dn) and 27-positively (27up) ID4-correlated genes identified in **c** and survival [distant-metastasis free survival, DMFS ($N = 275$); recurrence-free survival, RFS ($N = 417$)] in BLBC patients. Survival analyses were performed using the KMplot database (www.kmplot.com) considering the mean expression of the selected genes. Evaluation of combined signatures (up/dn) has been performed by setting the 19dn signature as "inverted" in the KM plotter. "Best cutoff" setting was used to split patients. Asterisks refer to logrank $p$ (**$p < 0.005$, ***$p < 0.0005$). **e** Kaplan–Meier analysis showing the association between the expression of the two groups of 19-negatively (19dn) and 27-positively (27up) ID4-correlated genes identified in **c** and survival [distant-metastasis free survival, DMFS ($N = 275$); recurrence-free survival, RFS ($N = 417$)] in BLBC patients. Survival analyses were performed using the KMplot database (www.kmplot.com) considering the mean expression of the selected genes and using the median value to split the patients. Asterisks refer to logrank $p$ (*$p < 0.05$, ***$p < 0.0005$).

carried out on QuantStudio5 Fast Sequence Detection Systems (Applied Biosystems, Carlsbad, CA, USA). For real-time PCR analysis of circ_0076611 in serum samples from BC patients and in RIP/ChIRP experiments, a linear preamplification of circ_0076611 using oligonucleotides divF and divR (Supplementary Data 1) and AmpliTaq Gold™ 360 Master Mix was performed, and, subsequently, 1 μl of this reaction was used to perform real-time PCR with TaqMan assay as above described. All primers and TaqMan probe were purchased from IdT (Integrated DNA Technologies) if not specified differently. Supplementary Data 1 includes the various primers used in PCR.

**ID4 knockout by CRISPR-Cas9 gene editing**. To generate stable ID4 knockout cell lines, we used the Alt-R CRISPR-Cas9 System (IDT). MDA-MB-468 and M6 cell lines were transfected with the RNP complex formed by the crRNA and tracrRNA duplexes (Alt-R CRISPR-Cas9 tracrRNA—ATTO 550 cat # 1075927) and the Cas9 enzyme (Alt-R S.p. HiFi Cas9 Nuclease V3 cat # 1081060), using Lipofectamine RNAiMAX (Thermo Fisher Scientific cat # 13778030) following manufacturer's instructions. Stable ID4 knockout cells were selected by screening monoclones by real-time PCR and western blot to analyze ID4 expression. The Alt-R CRISPR-Cas9 crRNAs used, are reported in Supplementary Data 1.

**Generation of circ_0076611 expression vector**. CIRC and CIRC#del expression vectors were cloned as follows. Genomic regions enclosing VEGFA exon 7 plus flanking intronic regions (introns 6 and 7) were amplified using upstream forward primer 5'-ttcttcaagctt(gcctgagactcacccttgtc)-3' and reverse primers 5'-ttcttcctcgag(caggcaggcagctagaaact)-3' (for CIRC vector) or 5'-ttcttcctcgag(gagcagaaggaccacaggac)-3' (for CIRC#del vector). HindIII and XhoI restriction endonucleases were used for cloning into pCDNA3.1 vector. The CIRC and CIRC#del vectors were used for transient and stable transfections in BC cell lines, using Lipofectamine 2000 Reagent (Invitrogen) according to manufacturer's instructions.

**Analysis of circ_0076611 by in situ hybridization**. For ISH analysis of circ_0076611, the BaseScope Technology (Bio-techne) was applied, following manufacturer's instructions. BaseScope™ Probe -BA-Hs-VEGFA-E7-circRNA (Cat. no. 724521) located in Homo sapiens VEGFA transcript variant 1 circRNA Exon7/Exon7_1715/1584 junction has been used to detect circ_0076611. The experiments were performed on a commercially available TMA (TMA #BR1301 from US Biomax, Inc) and in two cell lines: MDA-MB-468 and MCF10A. Cells were fixed with 3.7% formaldehyde in PBS and permeabilized with 70% ethanol for at least 30 min. Staining was visualized with an optical microscope, and circ_0076611 distribution was counted manually with the aid of ImageJ software. Microscope image evaluation was performed independently and in blinded manner by two investigators. Based on the numbers of circ_0076611 dots per field we considered as positive all the cases showing >1 dot/field, analyzing 4–5 fields/case (×100 magnification).

**Exosomes isolation and RNA extraction**. Exosomes were isolated from CM of MDA-MB-468, HCC1395 and OVCAR3 cells cultured in a 100 mm dish in the presence of 8 ml RPMI medium supplemented with 10% Exosome-depleted FBS (Euroclone, Cat. No. ECS8001). Specifically, 5 ml of 24 h CM from cells cultured at 60–70% confluence was used to isolate exosomes using the ExoQuick-TC ULTRA EV Isolation Kit (SBI, System Biosciences), following manufacturer's instructions. Exosomes' characterization was performed by western blot, using specific EXOAB primary antibodies from the kit (anti-HSP70, anti-CD9, anti-CD63). Extraction of total RNA from pelleted exosomes was carried out using TRIzol reagent (Thermo Fisher Scientific) following manufacturer's instructions. In total, 10 μg of Glycogen (Thermo Fisher Scientific) was added to enhance RNA recovery during the precipitation step. RNA concentration was measured using a NanoDrop 2000 (Nanodrop Technologies, Wilmington, DE, USA) and 200 ng of total RNA from exosomes were used for reverse transcription using SuperScript™ IV VILO™ Master Mix (Invitrogen), following manufacturer's instructions but performing the reverse transcription reaction at 65 °C. In total, 1 μl of cDNA was used to analyze circRNA expression in PCR and in real-time PCR.

**RNA extraction from serum**. Total RNA was isolated from the sera (100 μl) of women with established BC ($N = 89$) collected before and after a 3-months daily treatment with Metformin (500 mg day$^{-1}$) as agent with anticancer activity within a randomized clinical trial[31–33] using the MagMAX™ mirVana Total RNA Isolation kit following the manufacturer's instructions (Thermo Fisher). Two spike-in controls (cel-miR-39-3p and cel-miR-54-3p) (Thermo Fisher Scientific), added to sera during the RNA-binding beads step, were used to assess the extraction yields. The exogenous miRNAs were diluted at a final concentration of 0.002 fmole/μl and added at a consistent volume of 1.25 μl per 100 μl of biofluid. RNAs were eluted in nuclease-free water and frozen at −80 °C.

**RNA immunoprecipitation (RIP) on lysates from crosslinked cells**. Cells were crosslinked with 254-nm UV light 800 mJ/cm$^2$ (using 10-cm dish with 2.5 ml PBS 1X) or with formaldehyde (FA) solution (50 mM HEPES-KOH pH 7.5, 100 mM NaCl, 1 mM EDTA, 0.5 mM EGTA, 11% formaldehyde, ddH2O) 1% final concentration, 10 min, RT. Crosslinking with FA was stopped with 125 mM Glycine solution for 5 min at RT.

For the analysis of nuclear proteins (PTBP1, SRSF1, FBL) nuclei were isolated before resuspension in 200 μl lysis buffer (Tris-HCl pH 7.5 50 mM, EDTA 1 mM, SDS 0.5%, DTT 1 mM) for each planned immunoprecipitated (IP) sample; lysed samples were sonicated to obtain a smear of 300–1000 bp. Samples were diluted with 400 μl of correction buffer (NP-40, 0.625%, DOC, 0.312%, MgCl2, 5.6 mM, Tris-HCl pH 7.5, 47.5 mM, NaCl, 187.5 mM, glycerol, 12.5%, DTT 1 mM). IP was carried out overnight at +4 °C using 30 μl Dynabeads protein G (Thermo Fisher Scientific) previously coated with antibody. IP was followed by 5 washes in NT2 buffer (Tris-HCl pH 7.4 50 mM, NaCl 150 mM, MgCl2 1 mM, NP-40 0.05%). Beads were resuspended in 50 μl DNase buffer (NaCl 100 mM, NP-40 0.1%) plus 1 μl TURBO DNA-free™ (Ambion, Cat. No. AM1907) and incubated 30 min on a rotator (1000 rpm) at 37 °C. In total, 50 μl proteinase K buffer 2X (NaCl 100 mM, Tris-HCl pH 7.0 20 mM, EDTA 2 mM, SDS 0.5%) plus 5 μl pK Qiagen were added and incubated 30 min at 50 °C on a rotator (1000 rpm). Reverse crosslinkinking was performed 30 min at 70 °C on a rotator (1000 rpm) and samples were transferred to new tubes. RNA was extracted using 400 ul TRIzol, as from manufacturer's instructions, and adding 10 μg of glycogen at the precipitation step. For protein analysis by WB in RIP experiments, RNaseA was also added at the DNase treatment step and beads were directly resuspended in 40 μl Laemli solution 4X and boiled for 5 min at 95 °C. Antibodies anti-PTBP1 (Cell Signaling: #72669), anti-EIF4B (Cell Signaling: #3592), anti-EIF4G (Cell Signaling: #2469), anti-SRSF1 (Santa Cruz: SF2/ASF$^{96}$ sc-33652) and anti-Fibrillarin (Abcam: FBL ab5821) were used.

**ChIRP assay of circ_0076611, RNAseq and bioinformatic analysis**. ChIRP was performed as described[53,54]. Twenty million cells were used for each condition. The biotinylated oligonucleotides directed to circ_0076611 and to control LacZ are listed in Supplementary Data 1. Biotinylated oligonucleotides were recovered after hybridization step using Dynabeads® MyOne™ Streptavidin C1 (Invitrogen). After the washing steps, the whole recovered material was used for RNA purification with TRIzol. RNA was resuspended in 20 μl of H2O and 4 μl were used for reverse transcription using SuperScript™ IV VILO™ Master Mix (Invitrogen) to check enrichment of circ_0076611 and of its target RNAs.

RNAseq was performed on DNase-treated RNA recovered from pull-down experiments, and from input samples as well, on the Illumina platform (NextSeq instrument). The RNA from the pull-down experiments was controlled on a Bioanalyzer with the Agilent RNA 6000 Pico Kit (Agilent Technologies, Santa Clara, CA, USA) to verify the suitable size distribution and to simultaneously quantify the low RNA amounts. The RNA libraries for the RNA sequencing were prepared using the SMARTer Stranded Total RNA Sample Prep Kit—Low Input Mammalian (Takara Bio. USA, Inc). Each ChIRP-RNAseq was conducted combining two independent biological replicates. For the input we used 100 ng total RNA and performed ribosomal depletion with RiboGone (Takara Bio. USA, Inc) prior to cDNA and library preparation. For the RNA deriving from the ChIRP experiments we used 600 pg for cDNA and library preparation following the manufacturer's instructions. The quality of the resulting libraries was controlled on a Bioanalyzer using the High Sensitivity DNA Kit (Agilent Technologies, Santa Clara, CA, USA). The quantification of the libraries was performed by qPCR. Sequencing was carried out on a NextSeq 500 instrument (Illumina Inc., San Diego, CA, USA), sequencing in paired-end mode 76 bp.

Paired-end reads were aligned using STAR[55] version 2.7.3a to build version hg38 of the human genome. Reads aligning on ribosomal RNA were discarded using RSeQC. Strand specific coverage files (bigwig) were generated from the aligned reads using deepTools. Counts for GENCODE (version 35) annotated genes and VEGFA exons were calculated from the aligned reads using featureCounts function of the Rsubread R package[56] and R (version 3.6.3). For the analysis at gene level, counts for VEGFA were removed, then normalization and differential expression analysis were carried out using edgeR R package. Raw counts were normalized to obtain Counts Per Million mapped reads (CPM) and Fragments Per Kilobase per Million mapped reads (FPKM). Only genes with a CPM greater than 1 in at least one sample were retained for downstream analysis. For the analysis at VEGFA exon level, counts for VEGFA exon 7 were removed, then exon counts were divided by the total number of reads mapping on VEGFA. After removal of reads relative to rRNA and VEGFA we observed enrichment of 321 transcripts (FPKM (circ_0076611) > 15; FPKM (input) > 5 FC > 5; $p$ value < 0.05).

**Western blot**. For the Western blot analysis, cells were lysed in buffer with 50 mM Tris-HCl pH 7.6, 0.15 M NaCl, 5 mM EDTA, 1% Triton X- 100 and fresh protease inhibitors. Extracts were sonicated for 10 sec + 15 sec at 80% amplitude and centrifuged at ~12,000 rpm for 10 min to remove cell debris. The protein concentration was measured using a BCA protein assay kit (Thermo Scientific). The lysate was mixed with 4× Laemmli buffer and boiled for 5 min. Total protein extracts were resolved on polyacrylamide gel and then transferred onto nitrocellulose membrane. The following primary antibodies were used: DO-1 (anti-p53, Cell Signaling), B-5 (anti-ID4, Santa Cruz), ab46154 (anti-VEGFA, Abcam), abx129570 (anti-VEGF121, Abbexa), A301-985A100 (anti-BRD4, Bethyl), ab28283 (anti-CCND3, Abcam), SC-40 (anti- C-Myc, Santa Cruz), E1J4K (anti- C-Myc-p (Ser62), Cell Signaling), DM1A (anti-α-Tubulin, Cell Signaling), SC-47724 (anti-GAPDH, Santa Cruz). Secondary antibody fused with horseradish peroxidase was used for chemiluminescence detection on a UVITEC instrument (Uvitec, Cambridge, UK).

**Enzyme-linked immunosorbent assay**. Enzyme-linked immunosorbent assay kit was used according to the manufacturer's protocol for CXCL1/GRO alpha (4A-Biotech, CHE0065). The experiments were performed in MDA-MB-468 cell culture supernatants, after 48 h of transfection with si-circ_0076611, of which, the last 24 h in serum-free medium. Results were normalized to protein cell content.

**Polysome profiling and puromycin assay**. Cytoplasm fractionations on sucrose gradients were performed as follows. After 48 h of transfection with GapmeR to circ_0076611 or Negative Control, $20 \times 10^6$ cells were incubated 10 min with cyclo-heximide (Sigma-Aldrich) and then lysed with 400 μl of lysis buffer (10 mM Tris pH 7.5, 10 mM NaCl, 10 mM MgCl2, 1% Triton X-100) supplemented with 10 mM fresh DTT, 100 μg/ml cycloheximide, 1X PIC (Complete, EDTA free, Roche) and 1X RNase guard (Thermo Scientific). Cells were allowed to swell for 10 min on ice and the lysates were centrifuged for 10 min at 13,000 rpm at 4 °C. The supernatants were collected, loaded on 15–50% sucrose gradient and centrifuged at 38,000 rpm with a SW41 rotor (Beckman) for 1 h 30 min at 4 °C. Fractions were collected with a Biologic LP (Biorad). Then, 200 μl of each fraction were pooled together 3 by 3 obtaining four fractions (Heavy Polysomes, Light Polysomes, 40S/60S, Free RNA) and total

RNA was extracted using TRIzol (Invitrogen). Puromycin assay was performed in MDA-MB-468 cells transfected with GapmeR to circ_0076611 or Negative Control and treated with puromycin (#ant-pr-1, Invitrogen) at 10 ug/ml after 72 h of transfection. Incorporated puromycin protein level was then analyzed at 10 and 20 min after treatment by Western Blot analysis using mouse monoclonal anti-Puromycin, clone 12D10 (#MABE343, Millipore).

**Statistics and reproducibility.** For the statistical analysis of circ_0076611 level in serum samples, before and after treatment with Metformin, *p* value has been calculated using Wilcoxon matched-pairs signed rank test using the GraphPad Prism 7.0 software. For the majority of the experiments presented (RIP, ChIRP, RT-qPCR), significance was calculated using paired, two-tailed Student's *t* test using the GraphPad Prism 7.0 software, unless stated differently in the figure Legend, on at least three independent biological replicates of each experiment.

Analyses of associations between clinical variables by Chi-square test were performed using the GraphPad online 2 × 2 contingency table analysis tool (https://www.graphpad.com/quickcalcs/contingency1/).

Survival analyses of BLBC patients have been performed using the Kaplan–Meier plotter database (https://kmplot.com/analysis/index.php?p=service)[57], by selecting the basal subtype, based on the PAM50 classification. Mean expression of the multiple genes considered for the analysis was used. Patients were splitted on the median value of the signatures analyzed, except where indicated "best cut-off". Statistical analysis and graphs have been performed using the GraphPad Prism 7.0 software.

**Reporting summary.** Further information on research design is available in the Nature Research Reporting Summary linked to this article.

## Data availability

ChIRP-RNAseq data are available in GEO database as GSE183902. Uncropped blot/gel images are available as Supplementary Data 1. Source data for graphs is available in Figshare at https://figshare.com/s/7e6a0a9a0ea469d6d404. Plasmids for hsa_circ_0076611 expression are deposited at Addgene (Accession numbers 186343 and 186344).

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

## Acknowledgements

The authors received funding from AIRC (Associazione Italiana per la Ricerca sul Cancro) under IG 2018—ID. 21434 project—P.I. Fontemaggi Giulia and under 5xMille 2019–ID. 22759, awarded to G.B.

## Author contributions

Concept and design: G.F., F.F., and G.B. Acquisition, analysis, or interpretation of data: C.T., G.E., A.I., F.G., P.M., P.P., C.P., S.S., Z.I., M.F., T.D., E.G., L.P., and A.B. Drafting of the manuscript: G.F., G.B., and F.F. Critical revision of the manuscript for important intellectual content: G.F., C.T., A.I., F.F., A.F., and G.B. Statistical analysis: A.S. and M.F. Obtained funding: G.F. and G.B. All the authors have read the manuscript and approved its submission.

## Competing interests

The authors declare no competing interests.
