## [Peer Review File · Communications Biology]

Reviewers' comments:

Reviewer #1 (Remarks to the Author):

Alternative VEGFA splicing product, circ_0076611, is generated through the back splicing of exon 7 under the upregulation of the RNP complex and the inhibition of PTBP1 in the breast cancer cells. The circular VEGFA promotes cell proliferation in triple-negative breast cancer and maintains/increases the translation of the proliferation-related gene, VEGFA itself. The expression is significantly associated with tumor size and detectable in the exosomes of cells and serum from the patients. The work provides a potential therapeutic direction for TNBC by inhibiting VEGFA isoforms.

The authors perform a complete study of the circular VEGFA isoform from its gene to its phenotypic function. For example, to explore how circ_0076611 decreases the expression of proliferation-related genes, the authors first expelled the effects on the transcripts, then focused on the post-transcriptional aspects. Multiple methods are used to obtain conclusions. For instance, to verify that circ_0076611 expression is enhanced by MALAT1 and ID4, in MALAT1- or ID4-depleted cells, not only the expression of circ_0076611 is examined, but also the interaction between its target mRNA by RIP. To verify the inhibitory effect of PTBP1 on circ_0076611 expression, the authors do not only show the enhancement of circ_0076611 expression by PTBP1 inference but also show that the rescue of circ_0076611 expression by PTBP1 silencing in the MALAT1-depleted cells. The schematic figure 7J summarizes the upstream regulations of circ_0076611 production, which helps readers understand the work.

Despite these strengths, the paper has significant room for improvement. The impact is not clearly described, the manuscript could be improved by focusing on the impact and providing the corresponding data that explain that impact, rather than telling a story that is linear to the research team. Similarly, the structure of the paragraphs needs improvement. As written several paragraphs jump from topic to topic, presenting info but not connecting the importance of the information presented. This is especially necessary with the following paragraphs on p.4, starting with lines 118 and lines 134.

With regards to the results, a few questions need to be addressed. The data showing the circular RNA being present in only 1/2 of the samples makes one question the prevalence of this finding. More commentary and insight are needed into this in human samples. The scoring that is performed in Figure 2 is not clear, nor is the rationale for the scoring presented. While insight is given into what metformin does, it is not clear why metformin was used. It is not clear that the ChIRP assay can give function—rather where binding occurs---this result should be updated to be a bit more conservative regarding the findings.

The switch between primary tumor cells and cultured cells is not clearly delineated—additionally, it's not clear why the in vitro cells were chosen and only studied with some assays and not others. More insight into the larger goals of the work and why the models used were the right ones in answering the questions the investigators posed.

The conclusion from Figure 5G is not clear. It does not appear to follow that the translation (initiation) rate is decreased because both si-circ cells and control cells started polypeptide synthesis within 10 min and si-circ cells did not synthesize comparable levels of polypeptides in 40 min as the control did in 10 min. Please show that si-circ cells do not start synthesis when control cells have started to demonstrate the lower translation initiation rate OR si-circ cells can synthesize comparable levels of polypeptides within longer time points as the control does within shorter time points.

The results could be better contextualized in the discussion section. Much space is spent describing mechanisms that were not studied in the paper. The paper could be improved by spending this space to offer clear mechanistic connections to the field. Only a couple examples are offered, but this could be done throughout the discussion: *

- NRP and stemness and NRP does not clearly connect to the results presented. More insight should be offered regarding the connection that is being drawn.

- The discussion includes some conjecture on the role of inhibitors of SRPK1 that does not clearly connect to the work that was presented.

Methodology: While a strength of the paper is that several assays were performed, it is also a weakness because the reader is not brought along into the importance of the approaches used. For example, the manipulation and confirmation of circular RNA is not an area that most biomedical researchers are familiar. Moreover, it is not a common understanding of VEGF RNA. Language that explains how circular RNA is confirmed would go a long way to bringing the reader along the journey. Similarly, the concept of convergent vs divergent primers to confirm circular RNA is not a widely known. The development of the ISH using BaseScope technology could be very impactful, but as presented includes jargon that is not common in the VEGF field. Again, the paper could be improved throughout with citations and language that contextualize the significance of these approaches to the study.

Acronyms are used quite a bit in the paper and not always defined at the first use or at all. The readability of the paper would increase significantly, if fewer acronyms were used (possibly set a cut-off of how frequently they are used, and only use those that show up >n times), and they should be defined at the first use. A few examples are given for context, but others should be updated throughout the paper:

p.4 line 118: the MALAT1 was used previously without definition.

p. 36 Figure 2: Circ pos Circ neg (why not use the full terminology, there is space); T1-2, T3-4 (not clear what this means); T vs PT (why not say peritumor tissue in the figure)

Grammar needs to be improved in several portions of the paper. Only a few examples are given, but a reread of every line of the paper is encouraged

p.3, lines 108-111: This sentence required several re-reads to understand the key point.

Rephrasing or splitting into more than one sentence would help clarify this sentence.

p. 4, lines 122-124

p.3, lines 104-107

p.3, lines 114-117: This sentence is also very long and confusing, it's not clear how ID4 in BC cells reprograms TAMs---the mechanism should be more clearly described, the general description that is given is not clear.

The following are specific comments grouped by the portions of the manuscript. There are many minor issues with the usage of punctuations, articles, and prepositions. It is highly recommended to install an extension like "Grammarly" to the word processing software used. Such software will automatically offer this type of insight. Here, are several that were identified by, but it is strongly recommended that the authors perform a thorough grammar update.

Title:

1) "has_circ_0076611" is referred to as "circ_0076611" throughout the paper. Please be consistent.

2) Based on the results portion (line 545-578), the decreased translation rate is discussed in circ_0076611-depleted cells, rather than "translation initiation rate" as in the title. Please be consistent.

Abstract:

3) Please spell out the abbreviations, lncRNA MALAT1, ID4, SRSF1.

4) In the last sentence (line 65), please be specific about what functional aspects are involved.

Introduction:

5) VEGFA is indicated in line 91; while VEGF is mentioned as an example of the statement in line 94 and 96. Please be consistent.

6) It is written, in line 93, VEGFA is secreted through both paracrine and autocrine mechanisms; however, the first example corresponds to autocrine and the second one is paracrine. Please switch for consistency.

7) Please spell out the full names of MALAT1, SRSF1, ID4 when they first appear in the main body (line 110).

Materials and Methods:

8) Please expand the methods of cell transfection to offer greater detail.

9) It is more common to say "normalized to" than "normalized on" (line 245).

10) Change "hsa-circ_0076611" to "circ_0076611" to be consistent (line 253).

11) Add a comma between "For real-time PCR analysis of circ_0076611" and "primers"

div_F2/div_R2" (line 255-256).

- 12) Replace the comma with a semicolon (line 258).
- 13) Replace the second "using" with "and" (line 262).
- 14) Add a comma after "subsequently" (line 263).
- 15) Add a comma after "circ_0076611" (line 289).
- 16) Remove the first "in PCR" (line 317).
- 17) Expand the instruction for "(1000rpm)" as "on a rotator at 1000 rpm" (line 344 and 346).
- 18) Remove the last word "from" (line 348).
- 19) Different prepositions, "for" and "of", are used after "check enrichment" in parallel (line 363-364). Please keep consistent.
- 20) Replace the preposition "of" with "from" (line 367).
- 21) At least three replicates are required for the reproductivity of experiments. Please add at least one more independent replicate for CHIRP-RNAseq (line 372).
- 22) Remove the second "in" (line 396).
- 23) Supposed to be "10-15 s" and "~12,000 rpm" (line 398).
- 24) Replace the last preposition "into" with "onto" (line 401).
- 25) Add an "at" between "analyzed" and "10, 20, 40, and 60 min" (line 432).

Results:

- 26) Supposed to be "Figure 2F-G" (line 497) and "(Figure 2H-I)" (line 503).
- 27) The efficiency and specificity of circ_0076611 silencing are not examined to assure the oligonucleotides are functional and specific to circ_0076611. Please perform PCR or DNA electrophoresis to verify that. (line 519)
- 28) The efficiency and specificity of overexpression vector transfection are not examined. Please perform PCR or DNA electrophoresis. (line 525)
- 29) Please show the figures of western blot following polyribosome fractionation analysis (line 567) and RIP assay (line 572, 631-634), instead of just bar graphs.
- 30) Replace "to" with "in the" (line 571).
- 31) Please provide the rationale why the inverted repeated sequences for circ_0076611 production (line 604) corresponds to MALAT1 (line 608).
- 32) Add a comma after "To this end" (line 627).

Discussion:

- 33) Move "also" after "is" (line 675).
- 34) Add a comma after "moment" (line 676).
- 35) Rewrite the sentence (line 682-683)
- 36) Add a "the" before "absence" (line 686).
- 37) Replace "for" with "of" (line 689).
- 38) Add a comma after "circ_0076611".
- 39) Move "also" before "impacts" (line 743).

Reviewer #2 (Remarks to the Author):

In this study, Turco et al showed that circular isoform of VEGFA mRNA (circ_0076611), which is produced through the back splicing of exon-7, is associated with the size and pathogenesis of triple-negative breast tumors. circ_0076611 is produced through back splicing of exon-7 in breast cancer cells by a ribonucleoprotein complex comprising lncRNA-MALAT1, ID4 and SRSF1 and secreted through the exosomes to execute its function. This circ_0076611 binds to ribosomal RNAs finally impinging on translation efficiency and this circRNA controls a variety of target mRNAs including cell cycle regulators and VEGFA itself. The experimental design and the execution are excellent. The only thing the reviewer could not follow is how this circularRNA act in cancer cells in TME? Is it through autocrine or paracrine manner as it is secreted through exosome? It is very important to show that this circularRNA controls tumor growth in the rodent model.

Reviewer #3 (Remarks to the Author):

In this study, Turco C et al. identified circ_0076611, a circRNA from exon7 of VEGFA, involved in controlling the translation of target mRNAs to proteins in breast cancer. It was demonstrated that circ_0076611 could bind with EIF4B and facilitate the interaction of EIF4B and 5'UTR regions of

target mRNAs such as VEGFA, and consequently regulated their translation. A lncRNA MALAT1 regulated the biogenesis of circ_0076611 via forming a ribonucleoprotein complex with ID4 and splicing factor SRSF1. In addition, the ID4/circ_0076611 axis was suggested to predict the prognosis of the basal-like breast cancer. This study employed a combination of in vitro study and clinical analysis. Multiple techniques were used to prove the conclusion. It reached a very interesting area on how the circRNA biogenesis is regulated during it exerts function in tumorigenesis. Regulation of circ_0076611 by lncRNA MALAT1 is an interesting and unique part of this study. However, there are still some gaps when elaborating the mechanisms. And, some critical points need to be addressed for fully supporting the conclusion.

1. In Figure 1E, no error bars were presented.
2. In the line 497, Figure 4H-I didn't present the description here. In the line 503, "Figure 4J-K" might also be misplaced here.
3. In Figure 3F, it is shown that the expression of circ_0076611 was significantly inhibited by G-circ#1. However, the antisense oligonucleotide to block the circRNA also has partial complementary sequence with the linear VEGFA. What is the effect of G-circ#1 on the expression of linear VEGFA?
4. In Figure 3, it showed the circ_0076611 regulated cell proliferation. Does circ_0076611 only affect cell proliferation? Does it affect other features of breast cancer cells, like migration, invasion or survival?
5. In Figure 4B, no statistic labeling was found. In Figure 4C-E, should the data of circ-0076611 be compared with those of LacZ, acting as negative control, but not input?
6. In Figure 4I-L, it showed the si-circ#2 suppressed the protein levels of both CXCL16 and CXCL1 while only inhibited the mRNA expression of CXCL1 but not CXCL16. Can this difference be explained in detail?
7. In line 573 and Figure 5J, it was indicated that circ_0076611 may interact with EIF4B detected by RIP assay. However, it was not explained that why the EIF4B was selected. Was it from screening by protein identification analysis, such as Mass Spectrometry or from bioinformatics analysis? Were the other proteins related to translation initiation compared with EIF4B regarding the interaction with circ_0076611? The rationale for selecting EIF4B but not other translation initiation proteins is required.
8. It was mentioned in line 575-576 and Figure 5J that silencing circ_0076611 suppressed the interaction between EIF4B and the 5'-UTR regions of CCNB2 and VEGFA, which indicated the presence circ_0076611 facilitated the binding of EIF4B and 5'UTR regions. So, how circ_76611 functions as this facilitator? Does it also bind to the common sequence, such as Kozak consensus sequence, in 5'-UTR regions of different mRNAs?
9. It is better to explain in the Results section (line 584-588) that si-3 represent the depletion of MALAT1/ ID4/ mutant -53. It was mentioned in Figure legends but not explained very clearly in the Results. In addition, the sequence of si-3 was not clear listed in Table S1 with other siRNAs.
10. The statistic labeling was missing in Figure 7B right panel.
11. The bands should come from the same blot in Figure 7C lower panel showing SRSF1 protein levels with si-CTR and si-SRSF1.
12. In Figure 7D, did the experiments conducted in at least 3 repeat? Is there statistic significance between IgG and PTBP1 group?

Reviewer #1 (Remarks to the Author):

Alternative VEGFA splicing product, circ_0076611, is generated through the back splicing of exon 7 under the upregulation of the RNP complex and the inhibition of PTBP1 in the breast cancer cells. The circular VEGFA promotes cell proliferation in triple-negative breast cancer and maintains/increases the translation of the proliferation-related gene, VEGFA itself. The expression is significantly associated with tumor size and detectable in the exosomes of cells and serum from the patients. The work provides a potential therapeutic direction for TNBC by inhibiting VEGFA isoforms.

The authors perform a complete study of the circular VEGFA isoform from its gene to its phenotypic function. For example, to explore how circ_0076611 decreases the expression of proliferation-related genes, the authors first expelled the effects on the transcripts, then focused on the post-transcriptional aspects. Multiple methods are used to obtain conclusions. For instance, to verify that circ_0076611 expression is enhanced by MALAT1 and ID4, in MALAT1- or ID4-depleted cells, not only the expression of circ_0076611 is examined, but also the interaction between its target mRNA by RIP. To verify the inhibitory effect of PTBP1 on circ_0076611 expression, the authors do not only show the enhancement of circ_0076611 expression by PTBP1 inference but also show that the rescue of circ_0076611 expression by PTBP1 silencing in the MALAT1-depleted cells. The schematic figure 7J summarizes the upstream regulations of circ_0076611 production, which helps readers understand the work.

Despite these strengths, the paper has significant room for improvement. The impact is not clearly described, the manuscript could be improved by focusing on the impact and providing the corresponding data that explain that impact, rather than telling a story that is linear to the research team. Similarly, the structure of the paragraphs needs improvement. As written several paragraphs jump from topic to topic, presenting info but not connecting the importance of the information presented. This is especially necessary with the following paragraphs on p.4, starting with lines 118 and lines 134.

We thank the reviewer for all the suggestions, which allowed improvement of the study and of data presentation. We have extensively revised the manuscript trying to present the data as clearly as possible, highlighting the relevance of the findings. We have included (as manuscript related file) a version of the manuscript with all the text changes highlighted.

With regards to the results, a few questions need to be addressed. The data showing the circular RNA being present in only ½ of the samples makes one question the prevalence of this finding. More commentary and insight are needed into this in human samples. The scoring that is performed in Figure 2 is not clear, nor is the rationale for the scoring presented.

We thank the reviewer for this comment. It is true that we observed circ_0076611 expressed in half of the samples of the tissue microarray. However, this analysis has been performed using *in situ hybridization*, which is based on the use of a single probe pair covering the splicing junction of the circRNA. This method, despite highly specific, may have limited sensibility. On this basis we cannot exclude that a higher percentage of cases expresses the circRNA. It is also important to consider that circ_0076611 is controlled by ID4 and its expression is associated to ID4 protein in TNBC (Figure 8A). As ID4 protein is expressed in 70% of TNBC, with 30% of cases showing high score of ID4 expression, it is

possible that circ_0076611 is not ubiquitously expressed in TNBC because it depends on the presence of ID4.

This has been also included in the discussion (lines 723-726).

With regard to the scoring of circ_0076611 in ISH analysis, as in the previous version of the MS the scoring was arbitrary (due to the fact that the images obtained didn't allow an image-based quantitative software analysis), we have revised Figures 2 and 8, considering only positive vs. negative cases (Figure 2D, Figure 8A).

While insight is given into what metformin does, it is not clear why metformin was used. We apologize for lack of clarity in the presentation of rationale. We revised the presentation of the rationale in the results section (lines 526-536) and discussed more extensively the implications of the results in the discussion (lines 788-800).

It is not clear that the ChIRP assay can give function—rather where binding occurs—this result should be updated to be a bit more conservative regarding the findings.

We agree with the reviewer that ChIRP allows identification of mRNAs interacting with circ_0076611, not necessarily functional targets. We have modified the text and presented the data in a more conservative manner throughout the paragraph (from line 538).

The switch between primary tumor cells and cultured cells is not clearly delineated—additionally, it's not clear why the in vitro cells were chosen and only studied with some assays and not others. More insight into the larger goals of the work and why the models used were the right ones in answering the questions the investigators posed.

We apologize for the lack of clarity in indicating the choice of cell lines for the various experiments. We have now carefully revised the text providing explanation for the choice of the more suitable experimental systems for the various experiments.

Some examples of rationale for cell line choice:

- Nearly all the functional experiments (whole Figures 3, 4 and 5) were performed in MDA-MB-468 cells; this is a TNBC cell line extensively used in our previous work focused on the characterization of the linear VEGFA forms (doi: 10.15252/embr.201643370) and allowed us evaluating the circular VEGFA expression and activity in a system in which the VEGFA-related networks were already well characterized.
- The experiments presented in Figure 7, enclosing characterization of the protein complexes regulating the expression of circ_0076611, were performed using MDA-MB-468 and HCC1395, two cell lines chosen because they carry two hotspot mutation of TP53 with gain-of-function (p53R273H and p53R175H, respectively).
- The experiments in Figure 6 are carried out on various cell lines to strengthen as much as possible the concept that MALAT1 and ID4 control circ_0076611 expression.

The conclusion from Figure 5G is not clear. It does not appear to follow that the translation (initiation) rate is decreased because both si-circ cells and control cells started polypeptide synthesis within 10 min and si-circ cells did not synthesize comparable levels of

polypeptides in 40 min as the control did in 10 min. Please show that si-circ cells do not

start synthesis when control cells have started to demonstrate the lower translation initiation rate OR si-circ cells can synthesize comparable levels of polypeptides within longer time points as the control does within shorter time points.

We agree with the reviewer, and we apologize for the wrong presentation of data of previous Figure 5G. Indeed, in the previous analysis of the data the control (G-NC) was set to 1 in all the time points, confusing the results. Data enclosed in the new Figure 5G

(quantification on the left graph and representative blot on the right) show that at 10min control cells have started incorporation while si-circ incorporates very little, while at 20min si-circ has started the incorporation, comparable to the control-5min.

The results could be better contextualized in the discussion section. Much space is spent describing mechanisms that were not studied in the paper. The paper could be improved by spending this space to offer clear mechanistic connections to the field. Only a couple examples are offered, but this could be done throughout the discussion: *

- NRP and stemness and NRP does not clearly connect to the results presented. More insight should be offered regarding the connection that is being drawn.
- The discussion includes some conjecture on the role of inhibitors of SRPK1 that does not clearly connect to the work that was presented.

We thank the reviewer for this comment, which prompted us to extensively revise the discussion. We have eliminated many of the references that are not strictly related to our study and we expanded the discussion, trying as much as possible to link our data to the literature.

Methodology: While a strength of the paper is that several assays were performed, it is also a weakness because the reader is not brought along into the importance of the approaches used. For example,

- the manipulation and confirmation of circular RNA is not an area that most biomedical researchers are familiar. Moreover, it is not a common understanding of VEGF RNA. Language that explains how circular RNA is confirmed would go a long way to bringing the reader along the journey.
- Similarly, the concept of convergent vs divergent primers to confirm circular RNA is not a widely known.
- The development of the ISH using BaseScope technology could be very impactful, but as presented includes jargon that is not common in the VEGF field.

Again, the paper could be improved throughout with citations and language that contextualize the significance of these approaches to the study.

We have extensively revised the text, following the suggestions of the reviewer, to highlight the various experimental approaches and improve readability of the manuscript.

Acronyms are used quite a bit in the paper and not always defined at the first use or at all. The readability of the paper would increase significantly, if fewer acronyms were used (possibly set a cut-off of how frequently they are used, and only use those that show up >n times), and they should be defined at the first use. A few examples are given for context, but others should be updated throughout the paper:

p.4 line 118: the MALAT1 was used previously without definition.

p. 36 Figure 2: Circ pos Circ neg (why not use the full terminology, there is space); T1-2, T3-4 (not clear what this means); T vs PT (why not say peritumor tissue in the figure)

We have checked the manuscript for acronyms and tried to replace with full names, where possible.

Grammar needs to be improved in several portions of the paper. Only a few examples are given, but a reread of every line of the paper is encouraged

p.3, lines 108-111: This sentence required several re-reads to understand the key point. Rephrasing or splitting into more than one sentence would help clarify this sentence.

p. 4, lines 122-124

p.3, lines 104-107

p.3, lines 114-117: This sentence is also very long and confusing, it's not clear how ID4 in BC cells reprograms TAMs---the mechanism should be more clearly described, the general description that is given is not clear.

The following are specific comments grouped by the portions of the manuscript. There are many minor issues with the usage of punctuations, articles, and prepositions. It is highly recommended to install an extension like "Grammarly" to the word processing software used. Such software will automatically offer this type of insight. Here, are several that were identified by, but it is strongly recommended that the authors perform a thorough grammar update.

We have performed a thorough revision of the language, as suggested.

Title:

1) "has_circ_0076611" is referred to as "circ_oo76611" throughout the paper. Please be consistent.

We have checked consistency throughout the paper. In particular, we have mentioned hsa_circ_0076611 only at the very beginning and, later on, we have used the shortened name "circ_0076611".

2) Based on the results portion (line 545-578), the decreased translation rate is discussed in circ_0076611-depleted cells, rather than "translation initiation rate" as in the title. Please be consistent.

The title of this revised version of the manuscript has been changed to: "MALAT1-dependent hsa_circ_0076611 regulates translation rate in triple-negative breast cancer"

Abstract:

3) Please spell out the abbreviations, lncRNA MALAT1, ID4, SRSF1.

We have spelled out these abbreviations.

4) In the last sentence (line 65), please be specific about what functional aspects are involved.

We have revised the abstract.

Introduction:

5) VEGFA is indicated in line 91; while VEGF is mentioned as an example of the statement in line 94 and 96. Please be consistent.

We have checked consistency throughout the paper

6) It is written, in line 93, VEGFA is secreted through both paracrine and autocrine mechanisms; however, the first example corresponds to autocrine and the second one is paracrine. Please switch for consistency.

We have changed this, as suggested.

7) Please spell out the full names of MALAT1, SRSF1, ID4 when they first appear in the main body (line 110).

We have spelled out these abbreviations.

Materials and Methods:

8) Please expand the methods of cell transfection to offer greater detail.

We have expanded, as requested.

9) It is more common to say “normalized to” than “normalized on” (line 245).

We have changed this throughout the text.

10) Change “hsa-circ_0076611” to “circ_0076611” to be consistent (line 253).

We have changed this

11) Add a comma between “For real-time PCR analysis of circ_0076611” and “primers div_F2/div_R2” (line 255-256).

We have added this

12) Replace the comma with a semicolon (line 258).

This has been done

13) Replace the second “using” with “and” (line 262).

This has been done

14) Add a comma after “subsequently” (line 263).

This has been done

15) Add a comma after “circ_0076611” (line 289).

This has been done

16) Remove the first “in PCR” (line 317).

This has been done

17) Expand the instruction for “(1000rpm)” as “on a rotator at 1000 rpm” (line 344 and 346).

This has been done

18) Remove the last word “from” (line 348).

We have changed this

19) Different prepositions, “for” and “of”, are used after “check enrichment” in parallel (line 363-364). Please keep consistent.

We have changed/checked this throughout the text

20) Replace the preposition “of” with “from” (line 367).

This has been done

21) At least three replicates are required for the reproductivity of experiments. Please add at least one more independent replicate for ChIRP-RNAseq (line 372).

We apologize for the duplicate of the ChIRP-RNAseq experiment. However, all the validations have been performed on 3 additional independent biological replicates of the experiment, always confirming the RNAseq results.

22) Remove the second “in” (line 396).

This has been done

23) Supposed to be “10-15 s” and “~12,000 rpm” (line 398).

This has been changed

24) Replace the last preposition “into” with “onto” (line 401).

This has been done

25) Add an “at” between “analyzed” and “10, 20, 40, and 60 min” (line 432).

This has been done

Results:

26) Supposed to be “Figure 2F-G” (line 497) and “(Figure 2H-I)” (line 503).

This has been changed

27) The efficiency and specificity of circ_0076611 silencing are not examined to assure the oligonucleotides are functional and specific to circ_0076611. Please perform PCR or DNA

electrophoresis to verify that. (line 519). 28) The efficiency and specificity of overexpression vector transfection are not examined. Please perform PCR or DNA electrophoresis. (line 525)

The effect of the two oligonucleotides used in the study for silencing of circ_0076611 is presented in Figure 4G. We have evaluated here the effect of these oligonucleotides on circ_0076611 expression as well as on the expression of linear forms of VEGFA (VEGF165 and VEGF121). Moreover, we also evaluated the effect of circ_0076611 stable overexpression on circ_0076611 expression and on linear isoforms VEGF165 and VEGF121 and data have been included in Figure 3E.

29) Please show the figures of western blot following polyribosome fractionation analysis (line 567) and RIP assay (line 572, 631-634), instead of just bar graphs.

As the protein extracts obtained through polyribosome fractionation were not concentrated enough to allow western blot analysis, we have included, as additional control of the experiment, the analysis of the electrophoresis of the rRNA fractions obtained. Equal amounts of extracted RNAs from the various fraction pools (heavy, low, 40/60S, free) were evaluated using Agilent RNA 6000 Nano Kit on Bioanalyzer instrument. Electropherograms are shown in Figure 5I and show

comparable quality of fraction pools in G-NC vs. G-circ#1.

We included WB of RIP experiments for:
 EIF4B in Fig S3E-F
 EIF4G in Fig S3D
 PTBP1 in Fig 7H
 SRSF1 in Fig S4D

30) Replace “to” with “in the” (line 571).
 This has been changed

31) Please provide the rationale why the inverted repeated sequences for circ_0076611 production (line 604) corresponds to MALAT1 (line 608).
 The intronic sequences flanking exon7 contain consensus sequences for splicing factors SRSF1 and PTBP1, which are able to interact with pre-mRNA of circ_0076611 in RIP assays. MALAT1 probably regulates circ_0076611 by interacting with these splicing factors, impinging on exon 7 flanking regions. We have revised the whole paragraph to better explain these results (from line 634). Moreover, the results included in Figure 6 and 7 have been rearranged to optimize data presentation.

32) Add a comma after “To this end” (line 627).
 This has been done

Discussion:

- 33) Move “also” after “is” (line 675).
 - 34) Add a comma after “moment” (line 676).
 - 35) Rewrite the sentence (line 682-683)
 - 36) Add a “the” before “absence” (line 686).
 - 37) Replace “for” with “of” (line 689).
 - 38) Add a comma after “circ_0076611”.
 - 39) Move “also” before “impacts” (line 743).
- This has been done, as suggested.

Reviewer #2 (Remarks to the Author):

In this study, Turco et al showed that circular isoform of VEGFA mRNA (circ_0076611), which is produced through the back splicing of exon-7, is associated with the size and pathogenesis of triple-negative breast tumors. circ_0076611 is produced through back splicing of exon-7 in breast cancer cells by a ribonucleoprotein complex comprising lncRNA-MALAT1, ID4 and SRSF1 and secreted through the exosomes to execute its function. This circ_0076611 binds to ribosomal RNAs finally impinging on translation efficiency and this circRNA controls a variety of target mRNAs including cell cycle regulators and VEGFA itself. The experimental design and the execution are excellent. The only thing the reviewer could not follow is how this circular RNA act in cancer cells in TME? Is it through autocrine or paracrine manner as it is secreted through exosome? It is very important to show that this circular RNA controls tumor growth in the rodent model.

We thank the reviewer for his comments and suggestions. To address the issue raised by the reviewer, concerning the ability of circ_0076611 to exert autocrine or paracrine functions in the TME, we evaluated the proliferation of tumor cells (MDA-MB-468) and of untransformed mammary epithelial cells MCF10A, cultured with conditioned media (CM,

prepared in serum-free medium, see Methods) collected from control (EV, empty vector) or circ_0076611 overexpressing (circ o/e) MDA-MB-468 cells. As shown in the new Figure 3K, both MCF10A and MDA-MB-468 cells cultured with circ-o/e CM showed increased growth rate compared to EV-CM, suggesting both autocrine and paracrine activity. Colony-forming assays were also carried out in the same conditions using only MDA-MB-468, as MCF10A cells didn't form colonies in the serum-free-based CM. These

experiments confirmed an increase in the colony-forming activity of cells grown in presence of circ-o/e CM compared to EV CM (Figure 3L).

With regard to the in vivo evaluation of tumor growth, we were unable to perform these experiments due to a time issue; indeed, although we have submitted the animal testing protocol for ministerial approval, we have not yet received an approval. We hope to include these data in the study that further delves into the role of circ_0076611 in TME and which is still ongoing.

Reviewer #3 (Remarks to the Author):

In this study, Turco C et al. identified circ_0076611, a circRNA from exon7 of VEGFA, involved in controlling the translation of target mRNAs to proteins in breast cancer. It was demonstrated that circ_0076611 could bind with EIF4B and facilitate the interaction of EIF4B and 5'UTR regions of target mRNAs such as VEGFA, and consequently regulated their translation. A lncRNA MALAT1 regulated the biogenesis of circ_0076611 via forming a ribonucleoprotein complex with ID4 and splicing factor SRSF1. In addition, the ID4/circ_0076611 axis was suggested to predict the prognosis of the basal-like breast cancer. This study employed a combination of in vitro study and clinical analysis. Multiple techniques were used to prove the conclusion. It reached a very interesting area on how the circRNA biogenesis is regulated during it exerts function in tumorigenesis. Regulation of circ_0076611 by lncRNA MALAT1 is an interesting and unique part of this study. However, there are still some gaps when elaborating the mechanisms. And, some critical points need to be addressed for fully supporting the conclusion.

We thank the reviewer for all the suggestions, which allowed improvement of the manuscript and presentation of the data.

1. In Figure 1E, no error bars were presented.

We have included error bars.

2. In the line 497, Figure 4H-I didn't present the description here. In the line 503, "Figure 4J-K" might also be misplaced here.

We apologize for the mistakes. We have now indicated the right panels and included appropriate descriptions (from line 523).

3. In Figure 3F, it is shown that the expression of circ_0076611 was significantly inhibited by G-circ#1. However, the antisense oligonucleotide to block the circRNA also has partial complementary sequence with the linear VEGFA. What is the effect of G-circ#1 on the expression of linear VEGFA?

The effect of the two oligonucleotides used in the study for silencing of circ_0076611 on the expression of linear forms of VEGFA (VEGF165 and VEGF121) has been included in Figure 4G. Moreover, we also evaluated the effect of circ_0076611 stable overexpression on the same linear isoforms of VEGFA and data have been included in Figure 3E.

4. In Figure 3, it showed the circ_0076611 regulated cell proliferation. Does circ_0076611 only affect cell proliferation? Does it affect other features of breast cancer cells, like migration, invasion or survival?

We thank the reviewer for the suggestions. We have now included data relative to the impact of circ_0076611 on cell motility. Figure S2C-E shows that silencing of circ_0076611 significantly reduces invasion and migration in MDA-MB-231 cells, as assessed by transwell assay using matrigel-coated inserts and wound-healing assay. MDA-MB-231

cells have been used in these assays due to their high motility.

5. In Figure 4B, no statistic labeling was found. In Figure 4C-E, should the data of circ_0076611 be compared with those of LacZ, acting as negative control, but not input?

We apologize for presentation of inappropriate comparisons. ChIRP of circ_0076611 has now been moved to Figure 3A, including 3 biological replicates and statistical labeling.

Moreover, in ChIRP experiments of Figure 4 we have presented the comparison between circ_0076611 and LacZ, as suggested (see new analysis below).

6. In Figure 4I-L, it showed the si-circ#2 suppressed the protein levels of both CXCL16 and CXCL1 while only inhibited the mRNA expression of CXCL1 but not CXCL16. Can this difference be explained in detail?

We can hypothesize that CXCL1 might be subjected to different circ_0076611-dependent control layers, as for example: 1) direct binding and post-transcriptional regulation by

circ_0076611; 2) transcriptional regulation by some transcription factor whose expression is controlled by circ_0076611; indeed, among the circ_0076611-targets there are also various transcription factors, such as for example c-Myc and NF-YB (Figure 3C), that could control CXCL1 transcription. c-Myc, in particular was shown to induce Cxcl1 expression (DOI: 10.7554/eLife.50731), despite not in TNBC. We have included this possibility in the results (line 590).

7. In line 573 and Figure 5J, it was indicated that circ_0076611 may interact with EIF4B detected by RIP assay. However, it was not explained that why the EIF4B was selected. Was it from screening by protein identification analysis, such as Mass Spectrometry or from bioinformatics analysis? Were the other proteins related to translation initiation compared with EIF4B regarding the interaction with circ_0076611? The rationale for selecting EIF4B but not other translation initiation proteins is required.

We apologize for lack of clarity in the description of this figure. The observation that circ_0076611 mainly localizes in the fraction containing 40/60S ribosomal subunits suggested to us its possible involvement in the control of translation initiation. We then focused on proteins of the eIF4F cap-dependent complex (responsible for recognition of the mRNA and binding to the small ribosomal subunit) by testing initially eIF4B, a

component of eIF4F necessary for the RNA unwinding activity of the complex. We have now also included the evaluation of the interaction between circ_0076611 and eIF4G, an additional component of the eIF4F complex able to interact with poly(A) binding protein (PABP) and to circularize the RNA during translation; identification of the interaction between circ_0076611 and eIF4G further supported the involvement of this circRNA in translation initiation. These results have been included in the new Figure S3D. We have modified the text to better explain the rationale of these experiments (from line 618) and included a reference when we mentioned the eIF4F

complex.

8. It was mentioned in line 575-576 and Figure 5J that silencing circ_0076611 suppressed the interaction between EIF4B and the 5'-UTR regions of CCNB2 and VEGFA, which indicated the presence circ_0076611 facilitated the binding of EIF4B and 5'UTR regions. So, how circ_76611 functions as this facilitator? Does it also bind to the common sequence, such as Kozak consensus sequence, in 5'-UTR regions of different mRNAs?

We thank the reviewer for this comment. To address this issue and evaluate whether circ_0076611 could interact with motifs involved in translation initiation in the 5'-UTR

regions, we have analyzed the distribution of circ_0076611 ChIRP-RNAseq reads along the target mRNA gene body. The analysis of the mean enrichment profile (now included in Figure S1A-B) showed that the majority of reads fall in the middle regions of mRNAs, both considering the coding

sequence (-50/+50 relatively to start/stop codon, panel A) and the mRNA (-50/+50 relative to TSS/TES, panel B). This result suggests that circ_0076611 doesn't directly recognize translation initiation motifs; however, it could cause initiation factors to move closer to target mRNAs. This information has been included in the results (lines 545-546).

9. It is better to explain in the Results section (line 584-588) that si-3 represent the depletion of MALAT1/ ID4/ mutant -53. It was mentioned in Figure legends but not explained very clearly in the Results. In addition, the sequence of si-3 was not clear listed in Table S1 with other siRNAs.

We apologize for lack of clarity. We have now replaced "si-3" with si- MALAT1/ ID4/p53 in the Figure 6A, changed the text (line 639-640) and indicated the lacking siRNA/ASO sequences in the Table S1.

10. The statistic labeling was missing in Figure 7B right panel.

11. The bands should come from the same blot in Figure 7C lower panel showing SRSF1 protein levels with si-CTR and si-SRSF1.

We have replaced this panel with Figure 7D, adding biological replicates for statistical labeling.

We have replaced this with a new blot in Figure 7E.

12. In Figure 7D, did the experiments conducted in at least 3 repeat? Is there statistic significance between IgG and PTBP1 group?

We have performed additional biological replicates for statistical labeling and included this experiment in Figure 7G.

REVIEWERS' COMMENTS:

Reviewer #2 (Remarks to the Author):

The authors satisfactorily responded all the queries raised by the reviewer. The manuscript may be accepted for publication

Reviewer #3 (Remarks to the Author):

The authors made a significant effort in replying to my comments by providing additional new data. I agree to acceptance of this manuscript with comments of three minor corrections of the text and figure labeling.

1. Typo: "Fugure 2H" in line 527 should be "Figure 2H"
2. Figure S3D, the label of "eiF4G-Ab" should be "eIF4G-Ab"
3. EIF4B or eIF4B, it should be consistent in the text and figures.